# COMPOSERFLOW: STEP-BY-STEP COMPOSITIONAL SONG GENERATION

## ABSTRACT

Song generation models seek to produce audio recordings with vocals and instrumental accompaniment from user-provided lyrics and textual descriptions. While *end-to-end* approaches yield compelling results, they demand vast training data and computational resources. In this paper, we demonstrate that a *compositional* approach can make song generation far more data-efficient by decomposing the task into three sequential sub-tasks: melody composition, singing voice synthesis, and accompaniment generation. Although prior work exists for each sub-task, we show that naïvely chaining off-the-shelf models yields suboptimal outcomes. Instead, these components must be re-engineered with song generation in mind. To this end, we introduce *MIDI-informed* singing accompaniment generation—a novel technique unexplored in prior literature—that conditions accompaniment on MIDI representations of vocal melody, empirically boosting rhythmic and harmonic consistency between singing and instrumentation. By integrating pre-existing models with our newly trained components (requiring only 6k hours of audio data on a single RTX 3090 GPU), our pipeline achieves perceptual quality on par with leading end-to-end open-source models, while offering advantages in training efficiency, licensed singing voices from professional artists, and editable intermediates. We provide audio demos and will open-source our model at `https://composerflow.github.io/web/`.

## 1 INTRODUCTION

Traditional song production is a staged, collaborative workflow that progresses from songwriting (lyrics, melody, form), arrangement, multitrack recording, editing, mixing, and mastering. Each stage depends on specialized expertise—producers, topliners, instrumentalists, and engineers—and tight iterative feedback in a DAW, where decisions about key, tempo, harmony, vocal delivery, and sound design are refined across multiple passes. While this process is flexible and quality-driven, it is time- and resource-intensive; revisions at any stage often cascade into downstream rework. Producers preserve editability by saving stems and session states, yet precisely reproducing changes remains laborious. This conventional pipeline sets a high bar for control and fidelity—highlighting an opportunity for computational systems that retain its editability while reducing cost and turnaround.

Recent advances in large generative models have led to commercial systems (e.g., Suno (2025)) and emerging open-source models (Yuan et al., 2025; Liu et al., 2025b; Ning et al., 2025; Gong et al., 2025; Yang et al., 2025; Lei et al., 2025; Liu et al., 2025a) that demonstrate high-quality, convenient song generation. Diverging from the conventional studio workflow, these models accept lyrics and textual descriptions and produce songs in an **end-to-end** manner. However, this monolithic approach presents several challenges: (i) the lack of interpretable and editable intermediates makes refining unsatisfactory outputs difficult; (ii) directly learning a mapping from text to audio is notoriously data- and compute-hungry; and (iii) vocals may misalign with the lyrics and exhibit unnatural timbre.

A compelling, yet currently under-explored, alternative to these end-to-end systems is the **compositional** approach, which leverages a sequence of specialized component models to construct the final audio. Between the initial user input (lyrics/descriptions) and the final target audio, we identify three critical intermediate outputs: the *vocal MIDI score* (specifying pitch and duration for each syllable), the *singing voice audio* (rendering the melody and lyrics), and the *instrumental accompaniment audio* (forming the complete musical track). Song generation can thereby be decomposed into three

| System | Training data (hrs) | Training GPUs |
|---|---|---|
| YuE (Yuan et al., 2025) | 650K | $16\times$ H800 |
| DiffRhythm (Ning et al., 2025) | 60K | $8\times$ Huawei Ascend 910B |
| ACE-Step (Gong et al., 2025) | 100K | $120\times$ A100 |
| SongBloom (Yang et al., 2025) | 100K | $16\times$ A100 |
| JAM (Liu et al., 2025a) | 54K | $8\times$ H100 |
| Levo (Lei et al., 2025) | 110K | $8\times$ A100 |
| **Ours** | **6K** | **$1\times$ 3090** |

Table 1: Comparison of the proposed compositional pipeline with existing open-source song generation models, all of which are end-to-end models, in terms of data and computational demand.

sub-tasks: melody composition, singing voice synthesis (SVS), and singing accompaniment generation (SAG), each handled by a dedicated model. This modularity offers practical benefits, including editability of the intermediate stages (e.g., modifying the vocal MIDI score) and substantially reduced training costs for each component model compared to massive monolithic architectures.

While the compositional concept is not new, one might assume a functional system could be created by *chaining* off-the-shelf component models. Our literature review reveals **two discrepancies** that challenge this: first, there is little empirical data directly comparing the performance and resource efficiency of compositional methods against prevailing end-to-end architectures. Second, prior work treats these components *in isolation*, failing to exploit their interdependencies. We assert that achieving optimal data efficiency and perceptual quality requires more than simple concatenation—the components must be re-designed and integrated within a tightly-coupled pipeline.

Our core technical contribution is the introduction of *MIDI-informed* singing accompaniment generation (**MIDI-SAG**), a novel cross-stage conditioning mechanism previously unexplored. Existing (non-informed) accompaniment generation models (Donahue et al., 2023; Chen et al., 2024; Trinh et al., 2024) rely solely on the raw vocal audio as model input. In contrast, we exploit the fact that the underlying vocal melody is already available in a MIDI representation from the preceding stages, and use it as an additional condition for the accompaniment generator. Doing so boosts the *rhythmic* consistency between the singing and the accompaniment, as we can more easily trace the beats and downbeats from the MIDI input than from the vocal audio. Moreover, we can also improve *harmonic* consistency between vocal and backing by further integrating a *melody harmonization* module (Yi et al., 2022) that generates the chord progression that the accompaniment has to follow—this chord condition is not available directly from the vocal audio.

Besides, we address a practical limitation where conventional SAG models implicitly assume continuous vocal input. This assumption fails in real-world song generation, where the accompaniment component must generate the instrumental sections (e.g., intro, bridge, outro) despite the absence of vocal condition. While this challenge of **structural completeness** is naturally handled by end-to-end models, it presents a key concern for our compositional pipeline. We address this by leveraging the full-song rhythm, chord structure, and the inpainting and outpainting capabilities of latent diffusion (Tsai et al., 2025), ensuring seamless structural integrity throughout the generated song.

While the proposed idea is general, in our implementation we leverage the off-the-shelf model CSL-L2M (Chai & Wang, 2025) for melody composition, and train the SVS and SAG components on our own, as illustrated in Figure 1. For SVS, we adopt the FastSpeech architecture (Ren et al., 2020) with 10 hours of audio from two licensed professional singers. For MIDI-SAG, we curate 6k hours of pop song recordings, and adopt the approach of MuseControlLite (Tsai et al., 2025) to add time-varying controls (e.g., to handle the conditions from the MIDI) to Stable Audio Open (Evans et al., 2025). Model training are all accomplished using a single RTX 3090 GPU. The proposed pipeline employs dramatically fewer training resources compared to prevailing open-source architectures (see Table 1), yet our experiments demonstrate that it achieves comparable perceptual quality, while offering advantages in the use of licensed singing voices, better lyrics-to-vocal alignment, and editable intermediates. We provide extensive audio samples on our demo page, comparing our model's results against others and showcasing editability at different stages of the pipeline. We commit to open-sourcing our model upon publication.

Figure 1: Model overview. The proposed pipeline begins with lyrics and proceeds through lyrics-to-melody generation (through CSL-L2M), singing voice synthesis (SVS; through FastSpeech), melody harmonization (through Accomontage2), and the proposed MIDI-informed singing accompaniment generation (MIDI-SAG; through modifying MuseControlLite).

## 2 RELATED WORK

Musical audio generation has seen rapid progress in recent years. Specifically, text-to-music (TTM) generation models, such as MusicLM (Agostinelli et al., 2023), MusicGen (Copet et al., 2024) and Stable Audio Open (Evans et al., 2025), primarily focus on generating instrumental music from textual descriptions. In contrast, lyrics-to-song generation (or "song generation") models take lyrics as an additional input, aiming to produce full musical audio encompassing both singing vocals and instrumental accompaniment. OpenAI's Transformer-based Jukebox (Dhariwal et al., 2020) represents an early exemplar of this paradigm, followed by contemporaneous commercial systems such as Suno (2025), and a subsequent proliferation of open-source models, including Transformer-based ones such as YuE (Yuan et al., 2025) and Levo (Lei et al., 2025), and diffusion-based ones such as DiffRhythm (Ning et al., 2025), ACE-Step (Gong et al., 2025), and SongBloom (Yang et al., 2025). While these end-to-end approaches yield compelling results, their reliance on vast datasets (cf. Table 1) and extremely deep, non-modular architectures results in two constraints relevant to our work: high training cost, and a lack of interpretable or editable intermediate representations. These constraints motivate our investigation into a more resource-efficient, compositional alternative.

Long before the emergence of end-to-end song generation, extensive research existed for each sub-task within the compositional pipeline. This includes work on lyrics-to-melody generation (Yu et al., 2021; Sheng et al., 2021; Ju et al., 2021), SVS (Lu et al., 2020; Chen et al., 2020; Liu et al., 2022), and SAG (Donahue et al., 2023). However, most prior work tackles these specific sub-tasks in isolation, without implementing and comprehensively evaluating a unified, full-scale song generation pipeline. For instance, Melodist (Hong et al., 2024) focuses primarily on integrating SVS and SAG while assuming the melody and lyrics are pre-supplied, while SongComposer (Ding et al., 2025) concentrates mainly on lyrics-to-melody generation. Although models addressing these individual sub-tasks could theoretically be naïvely cascaded to realize a compositional song generation system, the performance, coherence, and efficiency of such an approach remain unexplored and unevaluated.

To our knowledge, existing SAG models (Donahue et al., 2023; Chen et al., 2024; Trinh et al., 2024) are designed for general applicability and solely condition on the raw vocal audio input, without assuming the availability of the underlying melody MIDI. This constraint presents a sub-optimality from the perspective of a compositional pipeline. Besides the MIDI-SAG idea, we also address the structural completeness issue—the need to generate musically relevant transitions and standalone instrumental passages guided by the broader context—which has not been tackled before.

## 3 COMPOSITIONAL SONG GENERATION FRAMEWORK

Formally, a compositional pipeline $P$ for song generation sequentially maps user input (lyrics $L$ and description $D$, both of which are sequences of words) to the final audio $A$ (a waveform encompassing vocal and backing) through three distinct, modular stages: $P = T_1 \circ T_2 \circ T_3$.

$$\{L, D\} \xrightarrow{T_1} M \xrightarrow{T_2} V \xrightarrow{T_3} A \,, \tag{1}$$

where $T_1$, $T_2$, and $T_3$ stand for the sub-tasks lyrics-to-melody generation, SVS, and (MIDI-)SAG, and $M$ and $V$ represent the intermediate outputs, the vocal melody MIDI and the vocal-only audio.

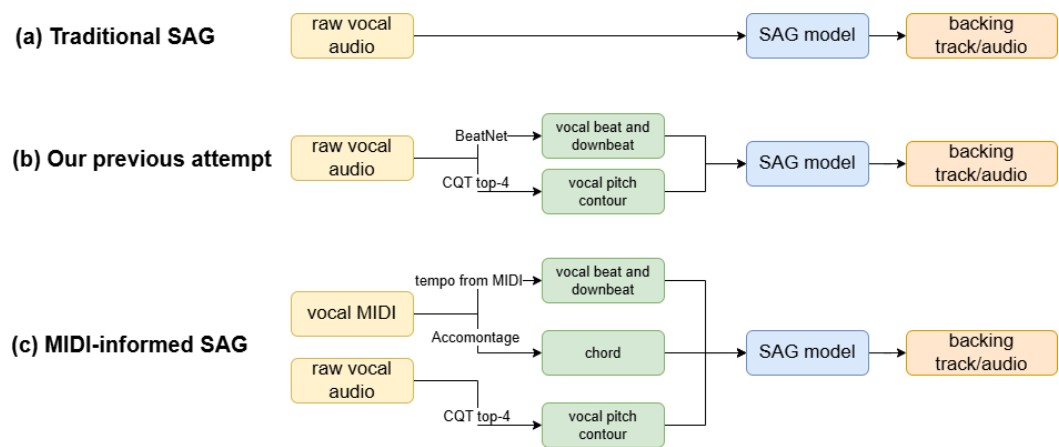

Figure 2: Illustration of conventional SAG versus MIDI-SAG; see Table 7 for empirical data.

Conventionally, SVS takes $L$ and $M$ as input and generates $V$, while SAG (Donahue et al., 2023) takes the vocal audio $V$ alone as input and generates $A$ (i.e., $V \to A$). In contrast, the proposed MIDI-SAG exploits the data coupling among all sub-tasks and learns the mapping $\{D, M, V\} \to A$.

Moreover, for stronger harmonic coherence between the vocal and backing, we insert an additional sub-task "melody harmonization" ($T_4$), that takes the symbolic vocal melody $M$ as input and generates a symbolic chord progression sequence $C$ (i.e., $T_4 : M \to C$) to guide the generation of the accompaniment. In consequence, our MIDI-SAG actually learns $\{D, M, C, V\} \to A$.

In what follows, we describe the core innovation MIDI-SAG ($T_3$) in greater details, and only provide high-level description for the others.

### 3.1 LYRICS-TO-MELODY GENERATION ($T_1$) AND SINGING VOICE SYNTHESIS ($T_2$)

We assume that the user has provided a full-song, sentence-by-sentence lyrics with structural indicators that divide the lyrics into sections (e.g., intro, verse, and chorus), but not sentence-level timestamps as required by some existing models such as DiffRhythm (Ning et al., 2025). Given the lyrics $L$, a lyrics-to-melody generation model ($T_1$) generates a sequence of monophonic musical notes in a symbolic, MIDI-like representation $M$ specifying the pitch and duration for each syllable in the lyrics. The task is challenging because lyrics and melody are only weakly correlated, but there are established models that we can leverage (e.g., (Chai & Wang, 2025; Ding et al., 2025)).

We note that melody composition is primarily *implicitly* performed by end-to-end song generation models, so there is no intermediate representation of the vocal melody $M$ that is interpretable and editable. In contrast, in our compositional pipeline, $M$ is the key piece of structured data that carries timing and pitch information required in all downstream sub-tasks $T_2$, $T_3$ and $T_4$.

SVS ($T_2$) is an established task leveraged within our compositional pipeline to address the common end-to-end issues of unnatural vocal timbre and high word error rate (Liu et al., 2025a). In our framework, any suitable SVS model can be used to render the vocal audio $V$ from the lyrics $L$ and melody $M$. Importantly, the melody $M$ from $T_1$ has to mark instrumental sections (e.g., intros) where the vocal output $V$ must be silent, ensuring synchronization with the broader song structure.

### 3.2 MIDI-INFORMED SINGING ACCOMPANIMENT GENERATION (MIDI-SAG) ($T_3$)

Given the vocal $V$, an SAG model generates the corresponding accompaniment that is supposedly rhythmically and harmonically coherent with the vocal, leading to the final mix $A$. This suggests that the beat and downbeat times of the vocal and back tracks have to be precisely synchronized, and that the vocal melody line must align with the backing track's underlying harmonic progression. Existing SAG models attempt to achieve this coherence in an end-to-end fashion (see Figure 4(a)), typically relying on massive training data (e.g., 46k hours of music for SingSong (Donahue et al., 2023) and 300k hours for FastSAG (Chen et al., 2024)). However, our pilot studies showed that this

approach yields insufficient coherence within our resource-constrained scenario (i.e., 6k hours of training data and one 3090 GPU), necessitating a form of explicit structural conditioning.

Our initial idea, as illustrated in Figure 4(b), is to extract the *vocal pitch contour* from the vocal audio $V$ as melody condition for harmonic improvement, and extract the *vocal beat and downbeat* from the vocal audio $V$ as rhythmic condition. However, our pilot studies suggested that this approach remains unreliable (see Table 7 for empirical data). The core difficulty lies in the fact that beat and downbeat detection from pure singing audio remains error-prone (Heydari et al., 2023).

The proposed MIDI-informed SAG directly exploits the inherent data coupling within our compositional pipeline by employing structural, time-varying conditions derived from the symbolic vocal melody $M$ (the output of $T_1$) to guide the accompaniment generation. As illustrated in Figure 4(c), this strategy offers two key advantages over audio-based methods:

- Enhanced rhythmic consistency: We extract precise beat and downbeat information readily available from the symbolic melody $M$, rather than relying on error-prone estimation from the vocal audio $V$. This foundational step significantly improves the reliability and accuracy of the rhythmic condition.

- Explicit harmonic guidance: We explicitly predict the accompaniment's chord progression $C$ from the symbolic vocal melody $M$ via an off-the-shelf melody harmonization model Yi et al. (2022). This process generates an explicit harmonic condition $C$ that accurately reflects the intended structure (for both vocal and instrumental sections), thereby greatly improving the accuracy of the harmonic guidance for the accompaniment generator.

Importantly, the structural information derived from MIDI-SAG extends its benefits to the overall song architecture. In addition to leveraging audio inpainting and outpainting capabilities of diffusion models for smooth transitions at section boundaries, our accompaniment generator has explicit rhythmic and harmonic conditions to follow even within the instrumental sections, i.e., when the vocal is absent. This ensures structural completeness throughout the entire composition, a guarantee that conventional SAG models are unable to provide.

## 4 IMPLEMENTATION

### 4.1 MODEL ARCHITECTURE AND TRAINING DATA FOR EACH SUB-TASK

As depicted in Figure 1, we employ two established external models to generate key intermediate symbolic representations. For lyrics-to-melody generation ($T_1$), we utilize CSL-L2M (Chai & Wang, 2025). This model supports conditioning on various attributes, including key, emotion, and reference MIDI. We exploit all these conditions via curating a bank of 1,000 reference attribute sets, to ensure that CSL-L2M produces reliable result. We also integrate Accomontage2 (Yi et al., 2022) for melody harmonization ($T_4$). We keep CSL-L2M and AccoMontage2 fixed, and train only the SVS and MIDI-SAG models. Please see the appendix (Sections B.1 and B.2) for details.

We note that the selection of specific model architectures throughout this paper is primarily for prototyping and validating the core idea of the compositional song generation framework. In principle, any other suitable model architecture could be used. However, the specific choices impose practical constraints; for instance, as CSL-L2M currently supports only Chinese lyrics, our implementation is consequently focused on Mandarin singers in SVS component and Mandarin pop for MIDI-SAG.

For SVS ($T_2$), we train FastSpeech (Ren et al., 2020) from scratch on an internal corpus totaling 10 hours from two licensed singers (male/female), with aligned MIDI notes, phoneme labels, and ground-truth melspectrograms. Training uses a single NVIDIA RTX 3090 for 24 hours. We then fine-tune a Parallel WaveGAN vocoder (Yamamoto et al., 2020) for one week on the same RTX 3090 to reconstruct waveform audio from melspectrograms. See Section B.3 for more details.

We build our MIDI-SAG model ($T_3$) based on fine-tuning the state-of-the-art latent diffusion-based TTM model Stable Audio Open (SAO) (Evans et al., 2025). SAO is naturally suited for generating the instrumental backing as it does not inherently synthesize singing voices. Our fine-tuning procedure is two-fold. First, we curate a set of 6,000 hours of Mandarin pop to fine-tune it, following the data preparation pipeline described in Section B.4. Second, we employ the lightweight mechanisms of MuseControlLite (Tsai et al., 2025) to add time-varying controls to SAO via adapters. Following

Table 2: Conditioning signals used during training and inference for MIDI-SAG.

| Feature | Training | Inference |
|---|---|---|
| Vocal pitch contour | CQT top-4 from vocal track (Hou et al., 2025) | CQT top-4 from vocal track (Hou et al., 2025) |
| Rhythm | Allin1 (Kim & Nam, 2023) | From the generated vocal MIDI score |
| Chord | chord detection (Park et al., 2019) | Accomontage2 (Yi et al., 2022) |
| Structure | Allin1 (Kim & Nam, 2023) | Provided by user |
| Key | Key CNN (Schreiber & Müller, 2019) | Provided by user or from generated vocal MIDI score |
| Reference audio | Randomly mask out one structure; the remaining structures are used as reference audio | From the previously generated song-structure segment |

Tsai et al. (2025), all time-varying conditions are temporally interpolated to a common sequence length and concatenated along the cross-attention feature dimension. MuseControlLite is trained using a single NVIDIA RTX 3090 with an equivalent batch size of 108, running for 9 days.

The original MuseControlLite supports two types of conditions—*audio conditions* that facilitates audio inpainting and outpainting, and *attribute conditions* for guiding melody, loudness, and rhythm. To adapt it to our song generation pipeline, we implement the following changes. First, to overcome the 47 s context limit of SAO, we revise and extend the audio continuation strategy of Tsai et al. (2025) to produce seamless long-form outputs. Second, unlike the original MuseControlLite, which exclusively fine-tunes the cross-attention layers, we additionally fine-tune selected self-attention blocks within SAO. This yields stable conditioning while allowing the partially unfrozen backbone to learn smoother transitions at section boundaries (see Table 8). Finally, we consider a richer set of time-varying conditions derived from the vocal melody MIDI, as detailed below.

### 4.2 Conditions for MIDI-SAG during Training and Inference Times

We consider a much more extensive set of controls than the original MuseControlLite (Tsai et al., 2025) to overcome the challenges presented in low-resource long-form song generation. At training time, some conditions can be extracted from the accompaniment for preciseness. However, at inference time, the conditions can only be computed from the vocal audio or symbolic vocal melody. We provide the details below and offer a summary in Table 2.

**Vocal pitch contour**. We first separate vocals and accompaniment with Mel-Band RoFormer (Wang et al., 2023), then select prominent pitches via the top-4 constant-Q-transform (CQT) method of Hou et al. (2025). During inference, we extract vocal pitch contour from the SVS-generated singing.

**Rhythm.** Our pilot study shows that existing beat tracking models do not work well for pure vocal audio. For example, BeatNet (Heydari et al., 2021) only obtains a Rhythm F1 score of 0.3449 according to our evaluation. Alternatively, at training time, we use All-In-One (Kim & Nam, 2023) to extract beat and downbeat timestamps from the backing audio, converting them into binary indicator sequences of shape $(T, 1)$, where $T$ is the number of time frames (1 if an event occurs at a given frame, 0 otherwise). A Gaussian filter is then applied to produce smooth "rhythm activation" curves. During inference, when All-In-One cannot be applied to singing voice, we instead derive beat and downbeat timings from the quantized MIDI generated by our lyrics-to-melody model (CSL-L2M (Chai & Wang, 2025)), which outputs melodies in 4/4 time.

**Chord.** Our preliminary experiments show that our SAG generates unstable harmony and weak progressions without the chord condition. To remedy this, at training time, we apply a chord detector (Park et al., 2019) to the separated backing track and encode the results as 12-bin chromagrams (pitch-class membership over time). During inference, chord sequences are provided by our melody harmonization model AccoMontage2 (Yi et al., 2022).

**Structure.** We extract section labels and timestamps with All-In-One (Kim & Nam, 2023), discarding truncated start/end fragments and retaining intro, outro, break, bridge, inst, solo, verse, chorus. Timestamps slice audio at section boundaries; section-level captions are produced by an large audio language model (LALM) AudioFlamingo3 (Goel et al., 2025), and the section labels themselves are

used as a time-varying conditioning signal. To avoid abrupt changes, we align boundaries with song structure transitions or downbeats during both training and inference times.

**Key.** While chords convey strong local tonality, we include a section-level key condition to capture broader tonal context. We apply key-CNN (Schreiber & Müller, 2019) per structure section, reflecting that key modulations often occur at section boundaries.

**Reference audio.** We use similar audio conditions as MuseControlLite to facilitate the use of inpainting and outpainting techniques to create the instrumental sections. We also introduce *backward* continuation: since intros lack vocal pitch contour and slightly degrade conditioning quality, when the target slice begins with an *intro*, we replace the reference with a *verse* section with a probability of 50% during training. During training, all songs are sliced at section boundaries: each slice begins at $S_i^{\text{start}}$, where $i \in \{$intro, verse, chorus, solo, inst, bridge, break, outro$\}$, and ends at $S_i^{\text{start}} + 47$. A 47-second window anchored at $S_i^{\text{start}}$ may span multiple structural sections. We use the audio from the first section as a reference and the time-varying conditions mentioned above to train the model.

We use all these conditions in the proposed pipeline, as illustrated in Figure 4 in the appendix. In our experiments, we provide ablation studies examining the importance of each condition.

### 4.3 INFERENCE PROCESS OF THE PROPOSED PIPELINE

As shown in Figure 1, users provide lyrics with structure tags, plus optional conditions (emotion and key). As described in Section 3.1, the system is conditioned on selected suitable statistical musical attributes, structure, and lyrics. The resulting melody and lyrics are then passed to the SVS model to synthesize a singing voice. Because the pitch contour of a quantized MIDI melody can deviate from the realized vocal pitch, we extract the vocal pitch contour from the synthesized singing voice and use it as a condition. In parallel, AccoMontage2 (Yi et al., 2022) performs melody harmonization to produce a chord progression with the generated vocal MIDI track. With the assembled controls—vocal melody, chords, rhythm (from MIDI timestamps), key (user-specified or inferred from the generated MIDI), and structure—MuseControlLite generates the backing track. Although MuseControlLite generates at most 47 s per pass, we extend duration via the audio-continuation procedure in Section 3.2: we first generate the *verse* without any audio condition, then generate the *intro*. Subsequent windows condition on the previously generated audio (anchored at the verse) and proceed section by section until the *outro*. Finally, we mix the completed backing track with the synthesized singing voice by summation and apply peak normalization to avoid clipping.

## 5 EXPERIMENTAL SETUP

### 5.1 BASELINES & EVALUATION DATASETS

For comparison, we select several representative baselines: Suno v4.5 (Suno, 2025), DiffRhythm 1.2-base (Ning et al., 2025), ACE-Step (Gong et al., 2025), and Levo (Lei et al., 2025). We exclude the following models from evaluation for fairness and comparability: SongGen (Liu et al., 2025b), whose outputs are limited to 30 s; SongBloom (Yang et al., 2025), which requires a 10 s reference audio as a style prompt (incompatible with our reference-free setting); and JAM (Liu et al., 2025a), which leverages phoneme-level timing supervision not available to other systems considered here.

All baselines and our model are evaluated on a 200-sample test set curated for this study. Each sample is automatically constructed via ChatGPT-5 system prompts that specify both lyrics and text prompts; the lyrics include one verse, one chorus, and one outro. We target 90–120 s *intro–verse–chorus–outro* forms, avoiding full-length pieces to mitigate listener fatigue. To accommodate heterogeneous model inputs, we also provide per-sample metadata (timestamps, timbre, genre, instruments) as separate annotations. For example, some systems (e.g., LeVo (Lei et al., 2025)) prefer tag-style textual descriptions, while others (e.g., DiffRhythm (Ning et al., 2025)) expect lyrics annotated with timestamps. We reformat inputs to match each model's specification while keeping the semantic content consistent across systems.

### 5.2 OBJECTIVE METRICS

Table 3: Objective evaluation for style alignment, inference speed, and phonme error rate.

| Model | CLAP ↑ | Inference time (s / song) ↓ | PER ↓ |
|---|---|---|---|
| Suno v4.5 (Suno, 2025) | **0.210** | — | 0.290 |
| ACE-Step (Gong et al., 2025) | 0.184 | 13.0 | 0.238 |
| DiffRhythm (Ning et al., 2025) | 0.187 | **11.5** | 0.325 |
| Levo (Lei et al., 2025) | 0.081 | 101.2 | 0.617 |
| **Ours** | 0.184 | 55.5 | 0.213 |

Table 4: Objective evaluation using Audiobox-Aesthetics and SongEval; the higher the better).

| Model | Audiobox | | | | SongEval | | | | |
|---|---|---|---|---|---|---|---|---|---|
| | CE | CU | PC | PQ | Coherence | Musicality | Memorability | Clarity | Naturalness |
| Suno v4.5 | 7.339 | 7.766 | 5.333 | 8.036 | **4.198** | **4.011** | **4.174** | **4.034** | **3.939** |
| ACE-Step | 7.209 | 7.642 | 5.820 | 7.948 | 3.449 | 3.214 | 3.203 | 3.216 | 3.162 |
| DiffRhythm | 7.530 | **7.791** | **6.336** | 8.189 | 3.740 | 3.419 | 3.595 | 3.512 | 3.354 |
| Levo | 7.565 | 7.674 | 4.993 | **8.295** | 3.392 | 3.272 | 3.198 | 3.265 | 3.155 |
| **Ours (w/ all conditions)** | **7.590** | 7.712 | 6.294 | 8.240 | 3.653 | 3.397 | 3.442 | 3.440 | 3.260 |
| Ours w/o chord | 7.480 | 7.643 | 6.324 | 8.170 | 3.472 | 3.271 | 3.263 | 3.267 | 3.123 |
| Ours w/o key | 7.591 | 7.722 | 6.206 | 8.244 | 3.634 | 3.387 | 3.436 | 3.426 | 3.251 |
| Ours w/o rhythm | 7.428 | 7.588 | 5.864 | 8.102 | 3.193 | 3.055 | 2.984 | 3.003 | 2.867 |
| Ours w/o structure | 7.589 | 7.715 | 6.287 | 8.231 | 3.660 | 3.423 | 3.468 | 3.450 | 3.290 |
| Ours w/o audio | 7.589 | 7.720 | 6.307 | 8.225 | 3.687 | 3.432 | 3.471 | 3.450 | 3.275 |
| Ours w/o vocal pitch contour | 6.148 | 6.787 | 5.728 | 7.408 | 2.607 | 2.460 | 2.430 | 2.399 | 2.396 |

To comprehensively evaluate the generated songs, we employ a set of objective metrics that capture different aspects of alignment and fidelity. These metrics quantify how well the outputs follow the intended textual and lyrical conditions. In addition, to better approximate human preferences, we further adopt automatic evaluation tools designed to reflect human-like judgments of aesthetics and production quality.

**Lyrics Alignment:** We employed Whisper ASR (Radford et al., 2022) to transcribe the generated vocals and compared the transcriptions with the ground-truth lyrics. Alignment quality is measured using the phoneme error rate (PER). We first convert both the predicted and reference texts into their phoneme representations. Then, PER is computed as $\text{PER} = \frac{S+D+I}{N}$, where $S$, $D$, and $I$ denote substitutions, deletions, and insertions, respectively. PER is the lower and the closer to 0 the better.

**Style Alignment:** We utilized CLAP (Wu et al., 2024b) to compute the cosine similarity between audio embeddings of the generated music and embeddings of the text prompts, quantifying adherence to the intended global style.

**Aesthetics Evaluation:** We used Audiobox-Aesthetics (Tjandra et al., 2025), to provide an automatic aesthetics qualification of the generated songs, capturing aspects such as clarity, richness, and technical fidelity. Specifically, Audiobox reports four sub-scores: Coherence (CE), Cultural Understanding (CU), Production Complexity (PC), and Production Quality (PQ).

**SongEval (Yao et al., 2025)**: A recently released evaluation tool specifically designed for songs, used to measure structural clarity, memorability, musical coherence, and overall musicality.

**Controllability**: To evaluate whether MIDI-SAG successfully aligns with the given conditions, we extract rhythm, chord, and key features from the generated backing audio using the same procedure as in training, except that we use BeatNet (Heydari et al., 2021) instead of All-in-one (Kim & Nam, 2023) to probe whether using different beat detection methods yield consistent results. Following Wu et al. (2024a), we use the F1 score to evaluate rhythm alignment, where the given timestamps and the detected timestamps are considered aligned if they differ by less than 70 milliseconds. For the chord condition, we also use the F1 score, computed between the chromagrams of the reference and generated audio. Key accuracy is defined straightforwardly: if the given key and the detected key match in both pitch class and mode (major/minor), the prediction is deemed correct.

## 5.3 Subjective metrics

Table 5: Subjective evaluation results across models; the higher the better.

| Model | Overall Preference | Lyrics Adherence | Musicality | Voice Naturalness | Song Structure Clarity |
|---|---|---|---|---|---|
| Suno v4.5 | **4.091** | **4.076** | **4.091** | **3.909** | **3.970** |
| ACE-Step | 2.803 | 3.455 | 3.045 | 2.561 | 3.045 |
| DiffRhythm | 2.409 | 2.788 | 2.758 | 2.561 | 2.530 |
| LeVo | 2.212 | 2.045 | 2.5 | 2.288 | 2.394 |
| **Ours** | 2.530 | 3.348 | 2.561 | 2.924 | 2.409 |

Table 6: Ablation study on conditioning signals.

| Setting | Chord F1 | Key Acc | Rhythm F1 |
|---|---|---|---|
| w/ all conditions | 0.9006 | 0.79 | 0.8339 |
| w/o chord | 0.3908 | 0.21 | 0.7870 |
| w/o key | 0.9027 | 0.78 | 0.8535 |
| w/o rhythm | 0.8914 | 0.79 | 0.4279 |
| w/o structure | 0.8957 | 0.79 | 0.8317 |
| w/o audio | **0.9027** | **0.84** | **0.8442** |
| w/o vocal pitch contour | 0.5930 | 0.69 | 0.4319 |

We conducted a mean opinion score (MOS)-style listening test with 33 anonymous participants recruited from the Internet, including self-reported music experts and non-experts. Each participant listened to 10 generated samples (5 models × 2 pairs of prompt-lyrics) and rated them on a scale of 1–5 (the higher the better) across the following five dimensions:

**Overall Preference:** Overall liking of the song.

**Lyrics Adherence:** Whether the vocal content matches the given lyrics.

**Musicality:** Perceived musical quality, including melody and harmony.

**Voice Naturalness:** Naturalness of the generated singing voice.

**Song Structure Clarity:** Whether the song structure clearly follows the provided lyrical format.

This evaluation design allows us to simultaneously assess text–music alignment, lyric–vocal alignment, and overall musical quality across systems.

## 6 EXPERIMENTAL RESULTS

### 6.1 PERFORMANCE COMPARISON WITH EXISTING MODELS

Table 3 presents the CLAP scores and PER of the evaluated models. We see that our model attains comparable CLAP scores as most open-source models, whereas the closed-source commercial system Suno has the highest CLAP, demonstrating superior text–audio alignment. In terms of PER, our model attains the lowest PER, slightly outperforming ACE-Step and Suno. We attribute this to the use of an dedicated SVS model capable of generating clear and temporally well-aligned vocals in our compositional approach.

Table 4 shows that, on Audiobox-Aesthetics, DiffRhythm scores the highest overall, with other models clustered closely behind. On the other hand, SongEval indicates that Suno is the strongest across perspectives, and the SongEval metrics are highly correlated to each other. Our model is slightly weaker than DiffRhythm but surpasses ACE-Step and LeVo.

Table 5 shows the subjective evaluation result. Consistent with the SongEval metrics, Suno outperforms all systems, highlighting a sizable gap between closed- and open-source models. Among open-source baselines, our system ranks above DiffRhythm and LeVo in *Overall Preference*, but trails behind ACE-Step. For *Lyrics Adherence*, our scores are on par with ACE-Step and exceed other open-source models; this partially diverges from the objective *Alignment* metric. We hypothesize that participants' adherence judgments are influenced by other perceptual factors, even though an SVS front-end should strongly align vocals to the provided lyrics. In *Musicality* and *Song Structure Clarity*, our model surpasses LeVo but falls short of the remaining systems. Notably, we obtain

Table 7: Rhythm and key alignment on a 200-song test set.

| Model / Setting | Rhythm F1 | Key acc. |
|---|---|---|
| Conventional SAG (cf. Figure 4(a)) | 0.64 | 0.55 |
| Modified SAG (cf. Figure 4(b)) | 0.55 | 0.72 |
| Proposed MIDI-SAG (cf. Figure 4(c)) | 0.81 | 0.90 |

Table 8: Modifying MuseControlLite for audio continuation

| MuseControlLite  (Tsai et al., 2025) | FD↓ | KL↓ | CLAP↑ | Smoothness value↑ |
|---|---|---|---|---|
| w/ audio condition | 111.58 | 0.2160 | 0.3622 | –0.2734 |
| w/ audio condition unfreeze self attnetion layers | 109.62 | 0.1794 | 0.3961 | –0.3529 |

the best *Voice Naturalness* among open-source methods, underscoring the benefit of using an SVS module within the compositional song generation pipeline.

## 6.2 ABLATION STUDY

Tables 4 and 6 present an ablation study clarifying the contribution of each conditioning signal to MIDI-SAG performance. We observe that condition alignment and SongEval scores are highest when MIDI-SAG is conditioned only on non-audio inputs, where the model is freed from reconciling with ambiguous reference audio, allowing it to strictly follow the remaining time-varying conditions. However, this simplification can introduce acoustic abruptness, which is easily detectable by listeners despite high SongEval scores. Further analysis reveals redundancy in harmonic conditions: removing the explicit key condition has negligible impact on key accuracy, and removing the chord condition leaves both chord and key alignment metrics high. This suggests that the explicit key condition is likely redundant. Conversely, excluding the vocal pitch contour leads to a performance drop in Chord F1, key accuracy, and rhythm F1, indicating that SAG relies heavily on this condition.

Table 7 empirically compares the three SAG settings discussed in Section 4, demonstrating the lack of rhythmic and harmonic consistency between vocal and backing using either the conventional SAG approach or the naïve modification detecting beats and downbeats directly from vocal audio. Finally, Table 8 demonstrates partially unfreezing self-attention layers of SAO improves result.

## 7 LIMITATIONS

Although our approach offers a promising alternative for song generation, several limitations remain. First, melodies produced by CSL-L2M (Chai & Wang, 2025) tend to degrade beyond $\sim 120\,\mathrm{s}$, likely because most of its training examples follow an intro–verse–chorus–outro form and are only $\sim 90$–$120\,\mathrm{s}$ long, complicating full-length generation. Second, our MuseControlLite-based SAG model often yields weaker intros than end-to-end baselines, leaving room for improvement.

## 8 CONCLUSION

In this paper, we proposed ComposerFlow, demonstrating that combining three lightweight modules (CSL-L2M (Chai & Wang, 2025), SVS, and Accomontage2 (Yi et al., 2022)) with a fine-tuned text-to-music model can effectively enable song generation while requiring significantly less training data and computational resources. Moreover, our model allows users to freely edit intermediate results—such as melodies and chords in MIDI format—and provides separate vocal and backing tracks for greater flexibility. Importantly, the modular design means that components can be replaced with newer models (e.g., more advanced SVS systems), as long as they satisfy the pipeline's input and output requirements. Thus, our model represents **a flexible paradigm rather than a fixed system**. We evaluated our model through both objective and subjective experiments and observed comparable results to state-of-the-art open-source systems. These findings suggest that generating high-quality songs does not necessarily require massive datasets, excessive computational power, or high energy consumption.

## 9 REPRODUCIBILITY STATEMENT

We will release the full training and inference code for our model, including the data-processing pipeline and configuration files. The base models are open sourced; we provide adapted implementations to meet our specific requirements. Implementation details are provided in Section 4.

## 10 ETHICS STATEMENT

This work develops a modular song-generation pipeline intended for research and creative assistance rather than full automation of musical labor. We mitigate risks as follows: (1) **Data and Copyright**: the SVS module is trained solely on licensed recordings from two professional singers; accompaniment modeling relies on time-varying symbolic controls (e.g., chords, rhythm, melody) rather than reproducing copyrighted waveforms, but was still trained to learn the underlying data distribution(2) **Environmental Impact**: training and fine-tuning are conducted on a single NVIDIA RTX 3090 GPU, substantially reducing compute and energy use relative to typical end-to-end systems. (3) **Responsible Use**: the system does not include identity cloning or voice-matching features and is intended for research, education, and human-in-the-loop creation. (4) **Transparency**: we acknowledge the use of large language models (LLMs) to polish writing and assist with code where appropriate. We will release code and configurations to support reproducibility and community scrutiny.

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

## A  USE OF LLMS

We use LLM to polish our paper, and parts of the code are collaborated with LLMs.

## B  IMPLEMENTATION DETAILS

### B.1  IMPLEMENTATION DETAILS ON LYRICS-TO-MELODY GENERATION

We adopt the state-of-the-art CSL-L2M model Chai & Wang (2025) to map input Chinese lyrics to a vocal MIDI score. CSL-L2M is a Transformer decoder with an in-attention mechanism Wu & Yang (2023) that supports fine-grained lyric–melody controls. We condition on the global key[1] and emotion[2], as well as sentence-level structure[3] and statistical musical attributes[4] that strengthen lyric–melody coupling. The statistical attributes are extracted from a vocal MIDI track. Users may specify emotion, key, and structure (the latter carried by the lyrics), and optionally provide a reference MIDI from which we derive the statistical attributes. The model performs best when the reference vocal MIDI track and input lyrics have similar section structure, line count, and per-line word counts.

To enable use without a user-provided MIDI, we curate a bank of 1,000 reference attribute sets. Given new lyrics, we score each candidate using a weighted sum

$$P \; = \; 0.4\,P_{\text{sent}} \; + \; 0.4\,P_{\text{prof}} \; + \; 0.2\,P_{\text{struct}},$$

where lower is better. Here, $P_{\text{sent}}$ penalizes differences in total line count (optionally rejecting candidates with fewer lines than the target); $P_{\text{prof}}$ is the mean absolute difference between per-line token counts—treating each visible Chinese character as one token—after padding the shorter sequence with its median and scaling by the maximum observed token count; and $P_{\text{struct}}$ compares section tags mapped to integers, counting position-wise mismatches and adding a penalty for extra sections, normalized by the longer sequence length.

### B.2  IMPLEMENTATION DETAILS ON MELODY HARMONIZATION

Because chord progressions are used as time-varying controls during training, inference must supply compatible chord sequences. We harmonize the vocal MIDI score produced by CSL-L2M (Chai & Wang, 2025) using AccoMontage2 (Yi et al., 2022). To provide an instrumental lead-in, we prepend a 4-bar intro without vocal and reuse (duplicate) the chord sequence from the first 4 melodic bars to harmonize this intro. Melody harmonization supplies chord progression, enabling the singing-accompaniment generator to produce coherent, musically appropriate harmony. If the automatically generated chords are unsatisfactory, users may instead provide their own or partial edit the generated progression.

### B.3  IMPLEMENTATION DETAILS ON SINGING VOICE SYNTHESIS

We add a MIDI-conditioning embedding that aligns each phoneme to its corresponding MIDI note. Before synthesis, we compare the register of the vocal MIDI track to typical vocal ranges—male (lower) vs. female (higher)—and test octave shifts $\Delta \in \{-12, 0, +12\}$ to align the vocal MIDI track with each profile's comfortable tessitura. We then choose the (singer, $\Delta$) that keeps the most notes inside the target range while minimizing $|\Delta|$.

### B.4  IMPLEMENTATION DETAILS ON SINGING ACCOMPANIMENT GENERATION

We curate a dataset to fine-tune Stable Audio Open by following the data preparation pipeline illustrated in Figure 3. Specifically, it separates vocal and backing track (Wang et al., 2023), captions the

---

[1] 12 major: C, D♭, D, E♭, E, F, F♯, G, A♭, A, B♭, B; 12 minor: c, c♯, d, d♯, e, f, f♯, g, g♯, a, b♭, b.

[2] Three emotions: Neutral, Positive, Negative.

[3] Five sections: Verse, Chorus, Insertion, Bridge, Outro.

[4] 12 attributes: pitch mean (PM), pitch variance (PV), pitch range (PR), direction of melodic motion (DMM), amount of arpeggiation (AA), chromatic motion (CM), duration mean (DM), duration variance (DV), duration range (DR), prevalence of most common note duration (MCD), note density (ND), fraction of lyric syllables to melody notes (Align).

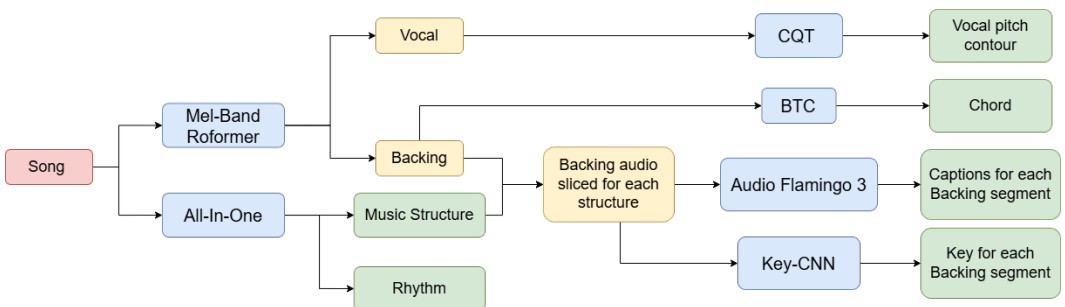

Figure 3: The data-preprocessing pipeline to curate data for fine-tuing Stable Audio Open to implement our MIDI-informed singing accompaniment generation (MIDI-SAG) model.

Table 9: Per-module inference latency for generating a 90–120 seconds *intro–verse–chorus–outro* song. For the singing-accompaniment stage, MuseControlLite (Tsai et al., 2025) is configured with 50 denoising steps.

| Module | Time (s) |
|---|---:|
| Lyrics-to-Melody | 10 |
| Melody Harmonization | 0.2 |
| Singing Voice Synthesis | 3 |
| singing accompaniment generation | 40 |

audio (Goel et al., 2025), extracts time-varying controls—chords (Park et al., 2019), rhythm (Kim & Nam, 2023), vocal pitch contour (Hou et al., 2025), structure tags (Kim & Nam, 2023), and local key (Schreiber & Müller, 2019) to serve as conditioning signals. We fetch the audio data of the Mandarin pop from the Internet, and keep it as an internal data for academic research purpose with no intention to distribute it further.

## C ADDITIONAL NOTES

To ensure that using Whisper is appropriate in our setting, we additionally compute the PER on real human singing from the CPOP (mir, 2018) dataset by comparing Whisper's transcriptions with the ground-truth lyrics. The resulting PER is 0.059, demonstrating that Whisper can effectively recognize singing voice.

Moreover, we observe that, for short lyric inputs, Suno v4.5 tends to repeat lines, which would otherwise inflates the PER. For a fair comparison, these repeated lines in the transcriptions are removed before computing the PER.

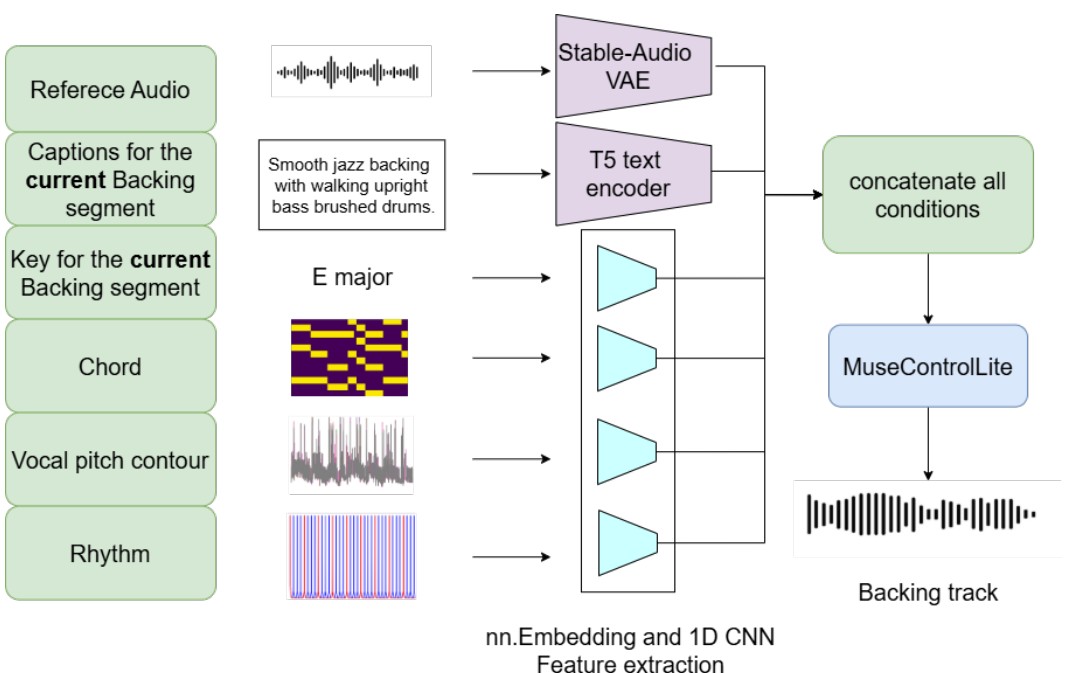

Figure 4: Equipping Stable-Audio Open with singing accompaniment generation and audio continuation abilities via MuseControlLite.

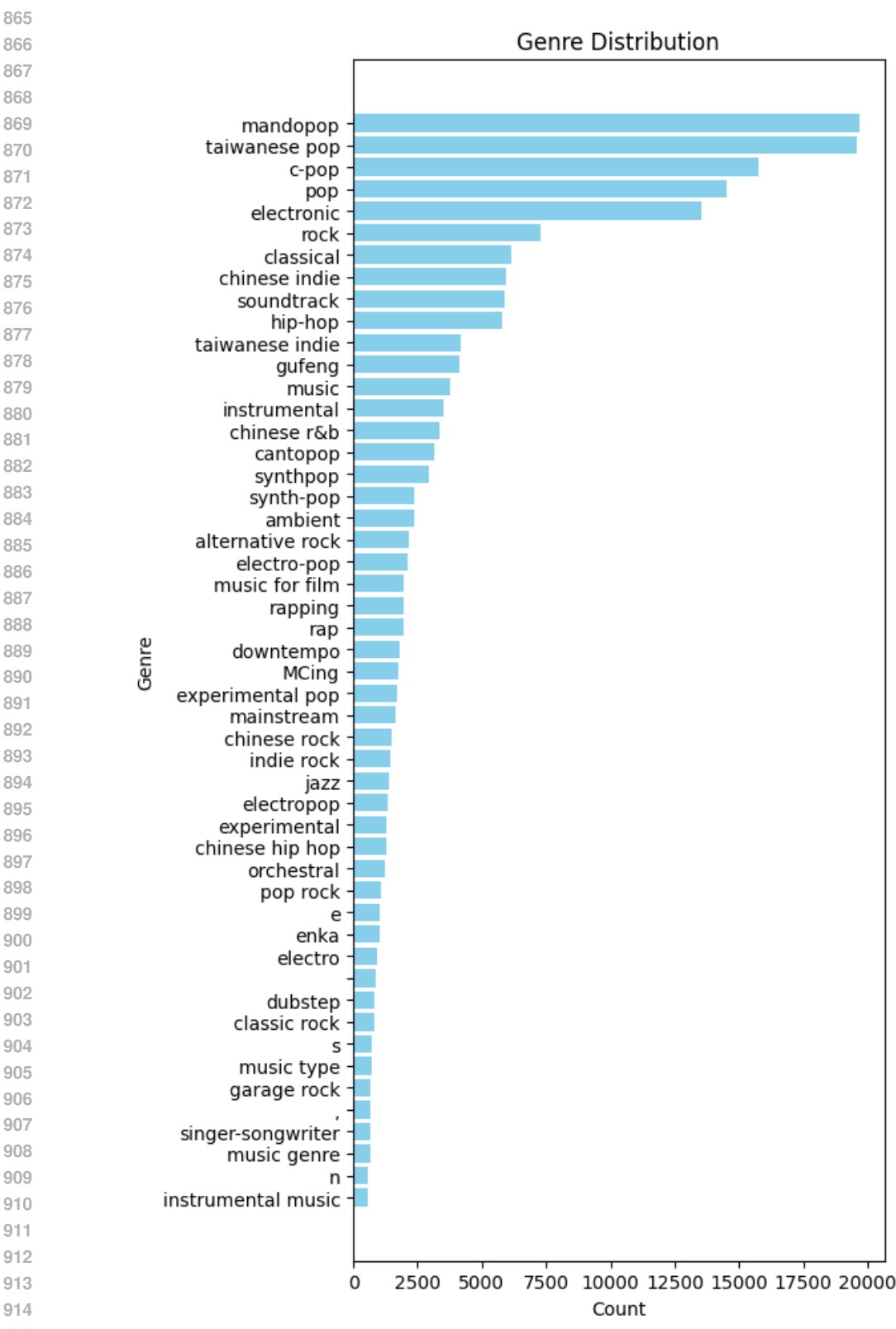

Figure 5: The genre distribution of the curated training dataset for training MIDI-SAG.

