# OpenReview forum: "ComposerFlow: Step-by-Step Compositional Song Generation"
_ICLR.cc/2026/Conference — Submitted to ICLR 2026_

### Official Review · Reviewer_TYbL · 2025-10-30

**Soundness:** 3
**Presentation:** 3
**Contribution:** 2
**Rating:** 2
**Confidence:** 5

**Summary:**

This paper introduces ComposerFlow, a modular pipeline for song generation that decomposes the process into four specialized stages: lyrics-to-melody, melody harmonization, singing-voice synthesis, and singing accompaniment generation. This approach reduces computational demands, requiring only a single RTX 3090 GPU and 6,000 hours of data for training, while enabling user edits to intermediate outputs like melodies and chords.

**Strengths:**

- This paper is original that it argues for a return to a structured, compositional pipeline reminiscent of traditional music production, while the field has converged on end-to-end models that map lyrics directly to audio. The key innovative step is the deliberate disassembly of the problem into four specialized sub-tasks (lyrics-to-melody, harmonization, SVS, SAG).
- This paper offers a solution to a critical weakness of end-to-end models: the lack of user control. By providing editable intermediate representations (MIDI melodies, chord progressions, separate vocal and backing tracks), it enables a "human-in-the-loop" creative process that is impossible with existing one-shot generators. This aligns much more closely with how music is actually composed and produced.

**Weaknesses:**

- This paper's core thesis is that a modular pipeline is superior to end-to-end models due to its editability and resource efficiency. While the resource efficiency is convincingly demonstrated, the ​claimed advantage of editability is not empirically validated.​​
  - This paper presents editability as a key benefit but provides no experiments or user studies showing that this leads to better outcomes or a more efficient creative process. It remains a theoretical feature. Can users actually achieve a superior final song through iterative edits? This paper doesn't show this.
  - As far as I listened in the demo page, perceivably, ComposerFlow is still way behind conventional end-to-end music generation models in terms of sound quality and musicality.
- The exclusion of models like SongBloom (for requiring a reference audio) seems reasonable for a "reference-free" test. However, excluding ​Jam​ for using "phoneme-level timing supervision not available to other systems" is quite vague. Excluding it makes the competitive landscape seem less advanced than it is.
- The most critical missing baseline is an ​ablated version of ComposerFlow itself. The paper does not show what happens if, for example, the structured controls (chords, rhythm, etc.) are removed from the SAG step, reverting to conditioning only on the raw vocal. While a "raw-vocal baseline" is mentioned in Section 3.3, its results are described qualitatively ("fails to reliably track tempo and key"). There are no quantitative results in Table 4 or 5 for this ablation.
- The pipeline inherits the constraints of its weakest component. The reliance on ​CSL-L2M for melody generation, which degrades beyond 120 seconds and is trained on a specific song structure (intro-verse-chorus-outro), means ComposerFlow is inherently limited to short, formulaic pop songs. It cannot generate more complex structures (e.g., verse-chorus-verse-bridge-chorus) or full-length compositions, which is a significant restriction. Furthermore, the SAG model's struggle with intros (due to lack of vocal conditioning) highlights a fragility in the pipeline's design.

**Questions:**

The paper highlights editability as a key advantage, but this is demonstrated only as a theoretical possibility. Could you provide any quantitative or qualitative evidence that this editability leads to a better or more efficient creative outcome?
For instance, did you perform any internal tests where you edited a melody or chord progression and measured the time/effort saved versus regenerating an entire song with an end-to-end model?

---

> ### Author Response · Authors · 2025-11-21
>
> Thank you for your comments!
>
> >This paper's core thesis is that a modular pipeline is superior to end-to-end models due to its editability and resource efficiency. While the resource efficiency is convincingly demonstrated, the claimed advantage of editability is not empirically validated.
>
> We acknowledge that the writing of the original paper does not effectively make the technical novelty of our work clear.  We have heavily rewritten our paper to have a focused presentation of the novel parts of our work.  In our revision, we have made it clear that the main novelty in this paper lies in the proposed staged pipeline and the MIDI-SAG.
>
> We are sorry for missing the editability evaluation. In our original submission, we thought editability is an inherent property or advantage of the proposed composable pipeline so we focused on evaluating the other aspects of the model. However, we agree that we should have at least demonstrated this capability. We have now updated the demo page (https://composerflow.github.io/web/) with:
> - Editing chord progressions and local text for each song structure segment.
> - Same lyrics and melody, but edit the key and singer
> - Same Lyrics and singer, different lyrics-to-melody model
> - Demonstrate the same lyrics, but with different melodies.
> Table 8 shows the rhythm, chord, key alignment between the given condition and the generated music, results indicate that the MIDI-SAG highly adheres the given condition, which means the minor edits of the intermidiate results (e.g. chords) will be reflected.
>
> >This paper presents editability as a key benefit but provides no experiments or user studies showing that this leads to better outcomes or a more efficient creative process. It remains a theoretical feature. Can users actually achieve a superior final song through iterative edits? This paper doesn't show this.
>
> We provide examples on the demo page showing that each intermediate results can be edited. Also, we evaluate the rhythm, chord, key alignment between the given condition and the generated music in Table 6. Results show that MIDI-SAG successfully follows the given condition, thus will reflect the edits in the final output. If the user desire specific melody, key, chord progressions, ConposerFlow is intuitively the best to choose, where end-to-end models lack such controls.
>
> >As far as I listened in the demo page, perceivably, ComposerFlow is still way behind conventional end-to-end music generation models in terms of sound quality and musicality.
>
> According to the overall preference in the subjective study, ComposerFlow is superior than DiffRhythm and Levo, but worse than Suno v4.5 and ACE-Step.
>
> >The exclusion of models like SongBloom (for requiring a reference audio) seems reasonable for a "reference-free" test. However, excluding Jam for using "phoneme-level timing supervision not available to other systems" is quite vague. Excluding it makes the competitive landscape seem less advanced than it is.
>
> We see your point. We excluded it as the word and phoneme-level timing control is absent for not only our model but also other models. We view the idea of using word and phoneme-level timing control of Jam as a complementary, generally-applicable technique that can be used to improve different models, and focus on prototyping and validating the effectiveness of the proposed compositional pipeline against end-to-end models.
>
> >The most critical missing baseline is an ablated version of ComposerFlow itself. The paper does not show what happens if, for example, the structured controls (chords, rhythm, etc.) are removed from the SAG step, reverting to conditioning only on the raw vocal. While a "raw-vocal baseline" is mentioned in Section 3.3, its results are described qualitatively ("fails to reliably track tempo and key"). There are no quantitative results in Table 4 or 5 for this ablation.
>
> We have updated Tables 4 and 6 with ablation study, demontrating the reliance of each condition. The results show that when the MIDI-SAG is conditioned only without audio, it performs the best regarding condition alignment and SongEval scores. When ignoring the reference audio, the model only need to follow other given time-varying conditions, however, there are some abrupt changes that are not captured by SongEval. Additionally, when excluding key condition, the key accuracy remains high; when excluding chord condition, both chord and key alignment remains high. As we know in music theory, the key is highly related to chord, thus showing the key condition might not be neccessary.

---

> ### Author Response · Authors · 2025-11-21
>
> >The pipeline inherits the constraints of its weakest component. The reliance on CSL-L2M for melody generation, which degrades beyond 120 seconds and is trained on a specific song structure (intro-verse-chorus-outro), means ComposerFlow is inherently limited to short, formulaic pop songs. It cannot generate more complex structures (e.g., verse-chorus-verse-bridge-chorus) or full-length compositions, which is a significant restriction. Furthermore, the SAG model's struggle with intros (due to lack of vocal conditioning) highlights a fragility in the pipeline's design.
>
>
> You are totally write that "The SAG model's struggle with intros (due to lack of vocal conditioning)"---that's exactly an issue the proposed MIDI-SAG mechanism is designed to address!
>
> We are sorry for not making this point clearer in the original version of the paper, but we have now made major revisions of the paper to make it clear how the proposed MIDI-SAG differs from conventional SAG approaches. In particular, we have added the following paragraph to describe how MIDI-SAG resolves the issue you nicely mentioned:
> ```
> Besides, we address a practical limitation where conventional SAG models implicitly assume continuous vocal input. This assumption fails in real-world song generation, where the accompaniment component must generate the instrumental sections (e.g., intro, bridge, outro) despite the absence of vocal conditioning. While this challenge of *structural completeness* is naturally handled by end-to-end models, it presents a key concern for our compositional pipeline. We address this by leveraging the full-song rhythm, chord structure, and the inpainting and outpainting capabilities of latent diffusion, ensuring seamless structural integrity throughout the generated song.
> ```
>
> Moreover, we note that the song structures (forms) supported by our current implementation are limited by the lyrics-to-melody model CSL-L2M, not by the SVS or the proposed MIDI-SAG model. When a full-song-level, multilingual lyrics–melody model becomes available, the MIDI-SAG model can accordingly perform full-song accompaniment generation.
>
>
> >The paper highlights editability as a key advantage, but this is demonstrated only as a theoretical possibility. Could you provide any quantitative or qualitative evidence that this editability leads to a better or more efficient creative outcome? For instance, did you perform any internal tests where you edited a melody or chord progression and measured the time/effort saved versus regenerating an entire song with an end-to-end model?
>
> We are sorry for missing this part. We thought editability is an inherent property or advantage of the proposed composable pipeline, so we focused on evaluating the other aspects of the model in our original submission. However, we agree that we should have at least demonstrated this capability. We have now updated the page ((https://composerflow.github.io/web/)) with:
> - Editing chord progressions and local text for each song structure segment.
> - Same lyrics and melody, but edit the key and singer
> - Same Lyrics and singer, different lyrics-to-melody model
> - Demonstrate the same lyrics, but with different melodies.
> Table 6 shows the rhythm, chord, and key alignment between the given condition and the generated music. The results indicate that the MIDI-SAG highly adheres to the given condition, which means the minor edits of the intermediate results (e.g., chords) will be reflected.
>
> The "time saved" reviewer mentioned could have a high variance among different users, and it is highly related to whether you desire a specific vocal midi track, chord progression, or singer identity. If the user has such a requirement, using a step-by-step generation pipeline would be a satisfying approach.

---

### Official Review · Reviewer_7aJG · 2025-10-30

**Soundness:** 2
**Presentation:** 3
**Contribution:** 2
**Rating:** 2
**Confidence:** 4

**Summary:**

This paper presents a multi-stage pipeline for song generation, which consists of four components: text-to-melody, melody harmonization, singing voice synthesis (SVS), and accompaniment generation. Among these, two modules are directly adopted from existing models, and the other two are fine-tuned based on publicly available systems.

**Strengths:**

1. The paper is generally easy to follow.
2. The overall pipeline is clearly described and logically structured.

**Weaknesses:**

1. The idea of a multi-stage pipeline for song generation was explored long before end-to-end text-to-song models. Frameworks combining lyrics-to-melody, SVS, and vocal-to-accompaniment stages have existed for years. Thus, the overall pipeline structure is not novel, and all the adopted models are based on existing methods. Both the novelty and technical contribution are quite limited.
2. The paper claims that, compared with end-to-end systems, the proposed multi-stage design is (1) resource-efficient and (2) allows users to revise intermediate results. However, these claims are not convincingly supported. Several critical issues specific to multi-stage systems remain unaddressed, and the results do not demonstrate clear advantages in these aspects.
  - Regarding resource efficiency: this mainly comes from using many existing pretrained models. However, it is not accurate to claim efficiency without accounting for the training costs of these reused models. Unless the total data and training time across all stages are demonstrably smaller than those required by end-to-end systems—and the quality remains competitive—the claim of efficiency is not well supported.
  - Regarding generation quality: multi-stage systems face inherent challenges. From a data perspective, they rely on existing weak models to parse songs into multiple components (e.g., structure, melody, chords) for training. Since these intermediate annotations depend on the accuracy of existing models, the resulting data are often noisy or unreliable, which constrains the overall performance. From a generation perspective, the overall performance is bounded by the capability of every component model within the pipeline. In the provided examples, the SVS component struggles with high notes and prosody, leading to suboptimal expressiveness and further affecting accompaniment generation. More importantly, the harmony between vocals and accompaniment is often poor, which can also be heard in the demos. This is a key challenge for multi-stage systems, yet the paper does not discuss or attempt to address it.
  - Regarding the second claimed advantage—“users can revise intermediate results”—the paper and demos do not provide any evidence or demonstration of this capability. It remains unclear how minor edits to intermediate outputs affect the final results, or whether such edits can preserve previously generated content while modifying only the relevant components. Moreover, several recent end-to-end systems already support controllable or editable inputs (e.g., SongEditor[1]), offering similar or better functionality with higher audio quality.

[1]Yang, Chenyu, et al. "SongEditor: Adapting Zero-Shot Song Generation Language Model as a Multi-Task Editor." Proceedings of the AAAI Conference on Artificial Intelligence. Vol. 39. No. 24. 2025.

**Questions:**

1. Please explain what specific efforts were made to address multi-stage pipeline challenges, such as stage-level suboptimality and vocal–accompaniment alignment, and how these compare with end-to-end models in terms of performance.
2. Please demonstrate the “revise intermediate results” capability and provide examples showing its effectiveness and advantages over existing systems.
3. Please clarify the quality and reliability of the annotated intermediate data used for training.

---

> ### Author Response · Authors · 2025-11-21
>
> Thank you for your comments!
>
> >The idea of a multi-stage pipeline for song generation was explored long before end-to-end text-to-song models. Frameworks combining lyrics-to-melody, SVS, and vocal-to-accompaniment stages have existed for years. Thus, the overall pipeline structure is not novel, and all the adopted models are based on existing methods. Both the novelty and technical contribution are quite limited.
>
> We acknowledge that the writing of the original paper does not effectively make the technical novelty of our work clear.  We have heavily rewritten our paper to have a focused presentation of the novel parts of our work.
> While the compositional concept is indeed not new, one might assume a functional system could be created by chaining off-the-shelf component models. Our literature review reveals two discrepancies that challenge this: first, there is little empirical data directly comparing the performance and resource
> efficiency of compositional methods against prevailing end-to-end architectures. Second, prior work treats these components in isolation, failing to exploit their interdependencies. We assert that achieving optimal data efficiency and perceptual quality requires more than simple concatenation—the components must be re-designed and integrated within a tightly-coupled pipeline.
> Our core technical contribution is the introduction of MIDI-informed singing accompaniment generation (MIDI-SAG), a novel cross-component conditioning mechanism previously unexplored. Existing (non-informed) accompaniment generation models (e.g., SingSong, FastSAG) rely solely on the raw vocal audio as model input. In contrast, we exploit the fact that the underlying vocal melody is already available in a MIDI representation from the preceding stages and use it as an additional condition for the accompaniment generator. Doing so boosts the rhythmic consistency between the singing and the accompaniment, as we can more easily trace the beats and downbeats from the MIDI input than from the vocal audio. Moreover, we can also improve harmonic consistency between vocal and backing by further integrating a melody harmonization module (Accomontage2) that generates the chord progression that the accompaniment has to follow—this chord condition is not available directly from the vocal audio.
>
> >The paper claims that, compared with end-to-end systems, the proposed multi-stage design is (1) resource-efficient and (2) allows users to revise intermediate results. However, these claims are not convincingly supported. Several critical issues specific to multi-stage systems remain unaddressed, and the results do not demonstrate clear advantages in these aspects.
>
> >Regarding resource efficiency: this mainly comes from using many existing pretrained models. However, it is not accurate to claim efficiency without accounting for the training costs of these reused models. Unless the total data and training time across all stages are demonstrably smaller than those required by end-to-end systems—and the quality remains competitive—the claim of efficiency is not well supported.
>
> The training data used for all modules are:
> - CSL-L2M: About 270 hours of lyrics-melody pairs.
> - SVS: 5 hours each for the two license data.
> - Accomontage2: No training data needed, pure template matching from its chord progression library.
> - MIDI-SAG: Approximately 5,600 hours of backing track audio.
> - Stable-Audio Open: 7300 hours
>
> Thus, the total training data for all models is approximately 13,180 hours, __still far less than other baselines__.
>
> Training time for each module in ComposerFlow:
> - CSL-L2M: single V100 1 day
> - SVS: single RTX 3090 1 day
> - Accomontage2: No training needed
> - MuseControlLite: single RTX 3090 9 days
> - Stable-Audio Open:
>     - For VAE: 456 hours × 32 A100s
>     - For DiT: 338 hours on ×64 A100s
>     - Total of 36224 A100 GPU hours
>
> Since the other baselines (i.e., DiffRhythm, Levo, Suno) didn't release the training time, we now compare with Acestep.
>
> Training time for each module in ACE-Step, for example:
> - Music DCAE: 120 hours × 120 A100s
> - Music DCAE: 264 hours × 120 A100s
> - Total of 46080 A100 GPU hours
>
> Comparing ComposerFlow with ACE-Step, the training time is still 20% less; however, the training time of ComposerFlow is mainly consumed by the Stable-Audio Open pretraining. ComposerFlow aims to utilize an open-source pretrained text-to-music model to adapt to MIDI-SAG. We consider __the training time of the text-to-music model can be ignored__ since it could be replaced by another open-source pretrained text-to-music model.
> If we really need to consider all the modules, the feature conditioners (e.g., T5, mT5 text encoders, lyrics encoders) should be counted in all end-to-end song generation models; however, the training time of these pretrained models that song generation models utilize is usually not reported.

---

> > ### Author Response · Authors · 2025-11-21
> >
> > >Regarding generation quality: multi-stage systems face inherent challenges. From a data perspective, they rely on existing weak models to parse songs into multiple components (e.g., structure, melody, chords) for training. Since these intermediate annotations depend on the accuracy of existing models, the resulting data are often noisy or unreliable, which constrains the overall performance. From a generation perspective, the overall performance is bounded by the capability of every component model within the pipeline.
> >
> > End-to-end models also partially rely on existing controls. For instance, Acestep uses structure condition from All-in-one, key (essentia), and bpm (Beatthis). While the papers of these music information retrieval papers have evaluate the effectiveness of their model, we would like to share another point of view: The ablation study showed in Table 6 use BeatNet rather than All-in-one to detect beat and downbeat, which means the condition use for inference MuseControlLite and the evaluation use different beat detection technique, but when all conitions are given, the Rhythm F1 score is still very high, showing that the both beat detection models may have a consensus. Additionally, as we know, the key is highly related to the chord progression. In Table 6, when the key condition is excluded, the model is still able to generate music with the same key accuracy as when all conditions are given, indicating that the model successfully follows the correct chord progression. However, when the chord condition is excluded, the model fails to follow the key condition, which indicates that the model fails to depend on the key condition, and the key accuracy is dependent on the chord condition.
> > We consider the extracted time-varying conditions (e.g., chord, rhythm) reliable.
> >
> > >In the provided examples, the SVS component struggles with high notes and prosody, leading to suboptimal expressiveness and further affecting accompaniment generation.
> >
> > Although we perform automatic pitch shifting as described in Section B.3, there might be some notes that are slightly out of the singer's comfortable tessitura, and this can be mitigated by manually changing the key, which is demonstrated in Section 3 in the demo page.
> >
> >
> > >More importantly, the harmony between vocals and accompaniment is often poor, which can also be heard in the demos. This is a key challenge for multi-stage systems, yet the paper does not discuss or attempt to address it.
> >
> > To our best knowledge, it remains difficult to specifically evaluate the harmony part, and this is an evaluation missing from all song generation papers.
> > However, an advantage of our pipeline is that, if a user is not satisfied with the harmony between the vocal and backing, the user can edit the backing track with different chord progressions, as shown in Section 2 of the demo page. We provide examples with the same lyrics and melody, but with different chord progressions.
> >
> > >Regarding the second claimed advantage—“users can revise intermediate results”—the paper and demos do not provide any evidence or demonstration of this capability. It remains unclear how minor edits to intermediate outputs affect the final results, or whether such edits can preserve previously generated content while modifying only the relevant components. Moreover, several recent end-to-end systems already support controllable or editable inputs (e.g., SongEditor[1]), offering similar or better functionality with higher audio quality.
> >
> > We are sorry for missing this part. We thought editability was an inherent property or advantage of the proposed composable pipeline, so we focused on evaluating the other aspects of the model in our original submission. However, we agree that we should have at least demonstrated this capability. We have now updated the demo page ((https://composerflow.github.io/web/)) with:
> > - Editing chord progressions and local text for each song structure segment.
> > - Same lyrics and melody, but edit the key and singer
> > - Same Lyrics and singer, different lyrics-to-melody model
> > - Demonstrate the same lyrics, but with different melodies.
> > Table 8 shows the rhythm, chord, and key alignment between the given condition and the generated music. The results indicate that the MIDI-SAG highly adheres to the given condition, which means the minor edits of the intermediate results (e.g., chords) will be reflected.

---

> > > ### Author Response · Authors · 2025-11-21
> > >
> > > >Please explain what specific efforts were made to address multi-stage pipeline challenges, such as stage-level suboptimality and vocal–accompaniment alignment, and how these compare with end-to-end models in terms of performance.
> > >
> > > Here are the efforts we made to prevent the cascading error:
> > > - Lyrics-to-melody: We set up a reference attribute set and a selection method described in Section 3.1. The reference statistical attributes will constrain the lyrics-to-melody generation, preventing awkward results.
> > > - Melody harmonization: Accomontage2 uses a chord progression matching paradigm to harmonize the given melody, and the chord templates from the chord library are either 4 bars or 8 bars, thus the chord progression itself will never be awkward.
> > > - Singing voice: We compare the register
> > > of the melody to typical vocal ranges—male (lower) vs. female (higher)—and test octave shifts∆ ∈ {−12, 0, +12} to align the melody with each profile’s comfortable tessitura. By doing these, we ensure the best quality of the singing voice.
> > > - Accompaniment generation: With the appropriate chord progression, singing voice, rhythm and other user given time-varying conditions, and the MIDI-SAG will generate backing track as expected.
> > >
> > > In our point of view, the cascading error seldom happens. The performance of the final output (song) are evaluated in Table 4 and 5, showing average performance among the baselines.
> > >
> > > >Please demonstrate the “revise intermediate results” capability and provide examples showing its effectiveness and advantages over existing systems.
> > >
> > > We update the demo page ((https://composerflow.github.io/web/)) with:
> > > - Editing chord progressions and local text for each song structure segment.
> > > - Same lyrics and melody, but edit the key and singer
> > > - Same Lyrics and singer, different lyrics-to-melody model
> > > - Demonstrate the same lyrics, but with different melodies.
> > > Table 6 shows the rhythm, chord, key alignment between the given condition and the generated music, results indicate that the MIDI-SAG highly adheres the given condition, which means the minor edits of the intermidiate results (e.g. chords) will be reflected.
> > >
> > > >Please clarify the quality and reliability of the annotated intermediate data used for training.
> > >
> > > For SVS, the training data is an internal corpus totaling 10 hours from two licensed singers (male/female), with aligned MIDI notes, phoneme labels, and ground-truth melspectrograms.
> > > For SAG, we fetch the audio data of the Mandarin pop from the Internet, and keep it as an internal data for academic research purpose with no intention to distribute it further.
> > > We have made such pieces of information clear in the revised paper.

---

### Official Review · Reviewer_yCCj · 2025-10-31

**Soundness:** 2
**Presentation:** 2
**Contribution:** 2
**Rating:** 2
**Confidence:** 4

**Summary:**

This paper presents ComposerFlow, a modular, step-by-step pipeline for generating songs from lyrics. ComposerFlow decomposes the task into a series of cascaded, specialized components: (1) lyrics-to-melody generation, (2) melody harmonization to create chords, (3) singing voice synthesis, and (4) singing accompaniment generation. The authors argue that this modular design offers greater editability, control, and significantly lower computational and data requirements. The paper evaluates ComposerFlow against several end-to-end baselines, including the commercial system Suno, using both objective and subjective metrics, and reports competitive performance with open-source models, particularly in voice naturalness and lyric alignment.

**Strengths:**

1. The core idea of a modular, controllable pipeline is well-motivated and addresses clear weaknesses in current end-to-end models (lack of editability, massive resource requirements, poor lyric alignment). This approach mirrors the traditional music production workflow and provides a practical alternative for researchers and creators without access to massive compute clusters.

2. The paper's claim of training the necessary components on a single consumer-grade GPU with a relatively small dataset is a significant strength. This "eco-friendly" approach makes the research more accessible, reproducible, and stands in stark contrast to the trend of ever-larger models, which is a valuable contribution to the community.

3. The modular design inherently allows users to intervene at intermediate stages (e.g., editing the MIDI melody, changing the chord progression, swapping the singer's voice). This is a crucial feature for creative applications and a major advantage over black-box end-to-end systems.

**Weaknesses:**

1. The primary weakness of this work is its lack of significant technical innovation. The paper is essentially an engineering effort that chains together existing, off-the-shelf models. While the way these components are connected and the data preprocessing pipeline (Figure 2) are non-trivial, the core generative models (CSL-L2M, AccoMontage2, FastSpeech, MuseControlLite) are all adopted from prior work. The main "novelty" lies in the application of MuseControlLite for the SAG task with structured controls, but even this is an adaptation of existing techniques. This feels more like a system demonstration paper than a research paper presenting a new fundamental method or discovery.

2. A well-known issue with cascaded pipelines is the problem of cascading errors. An awkward melody from the first stage will lead to poor chords, which in turn will result in a subpar accompaniment. The paper does not address how ComposerFlow mitigates this. Furthermore, by breaking the process into isolated steps, the system loses the ability to reason about the song holistically. For example, the accompaniment model has no direct knowledge of the lyrics' sentiment, only the derived melody and chords. An end-to-end model, in theory, can learn these complex, high-level dependencies between lyrics, melody, harmony, and instrumentation. This work fails to discuss this fundamental trade-off.

3. The evaluation, while extensive, has some notable issues: Comparing a Pipeline to End-to-End Models: It is fundamentally difficult to make a fair comparison between a modular system and a fully end-to-end one. ComposerFlow's superior lyric alignment (Table 4) is almost a given, as it uses a dedicated SVS module that is explicitly designed for this. This isn't a fair "win" but rather a direct consequence of the system's architecture.

4. The subjective results (Table 6) show that while ComposerFlow is competitive among open-source models, it is significantly outperformed by the commercial system Suno across all metrics. Even among open-source models, it does not consistently lead, trailing AceStep in Overall Preference and Musicality. Its main strength is Voice Naturalness, which is again expected due to the dedicated SVS module. Overall, the generated music quality appears to be average at best.

5. The reported inference time of 55.5s (Table 4) is not particularly fast, especially compared to AceStep (13.0s) and DiffRhythm (11.5s). While much faster than autoregressive models like Levo, it doesn't represent a clear advantage in efficiency at inference time.

6. The system is trained on Mandarin pop and evaluated on lyrics generated by ChatGPT. It is unclear how well this pipeline would generalize to other languages, genres (e.g., rap, where rhythm is paramount), or more poetic and structurally complex lyrics. The reliance on licensed data for the SVS, while ethically sound, also limits the vocal diversity of the system.

7. The figure quality is not good.

**Questions:**

1. The issue of cascading errors is critical in a pipeline like this. Did you observe instances where a poor output from an early module (e.g., a monotonous melody from CSL-L2M) led to a complete failure in the final output? Are there any mechanisms in place, or that you envision, to handle this, such as a feedback loop or a joint optimization strategy?

2. The SAG module is conditioned on several derived features (melody, chords, rhythm) but not directly on the raw vocals or lyrics. Do you think this limits the expressiveness of the accompaniment? For example, could an end-to-end model generate a more "empathetic" backing track that responds to the semantic meaning of the lyrics, which your system cannot?

3. In the subjective evaluation, what was the feedback from the "music experts"? Did they comment on the musical coherence and structure of the generated songs? The objective SongEval scores (Table 5) for ComposerFlow are decent but not outstanding. Does this align with expert opinion on aspects like harmonic progression and musical tension/release?

4. You mention that the pipeline is a "flexible paradigm rather than a fixed system." Could you elaborate on the challenges of swapping out a component? For example, if you were to replace the CSL-L2M melody generator with a different model that produces melodies in a different style, how much of the downstream pipeline (harmonization, accompaniment) would need to be re-tuned or re-trained to maintain quality?

5. Why was MuseControlLite chosen for the SAG task over other controllable generation models? And could you provide more detail on the decision to unfreeze the self-attention blocks of the Stable-Audio-Open backbone, and what effect this had on generation quality and continuity?

---

> ### Author Response · Authors · 2025-11-21
>
> Thank you for your comments!
>
> >The primary weakness of this work is its lack of significant technical innovation. The paper is essentially an engineering effort that chains together existing, off-the-shelf models. While the way these components are connected and the data preprocessing pipeline (Figure 2) are non-trivial, the core generative models (CSL-L2M, AccoMontage2, FastSpeech, MuseControlLite) are all adopted from prior work. The main "novelty" lies in the application of MuseControlLite for the SAG task with structured controls, but even this is an adaptation of existing techniques. This feels more like a system demonstration paper than a research paper presenting a new fundamental method or discovery.
>
> We acknowledge that the writing of the original paper does not effectively make the technical novelty of our work clear.  We have heavily rewritten our paper to have a focused presentation of the novel parts of our work.  In our revision, we argued that the novelty of this paper lies in the __proposed editable pipeline__ and the __MIDI-informed SAG__ method. In the revised paper, we also empirically demonstrate that naively chaining off-the-shelf models and using traditional SAG methods will not work, due to the fact that they cannot function properly when the vocal is silent. We present a fine-tuning method with a pretrained text-to-music model, since following explicit conditions is far more efficient than requiring it to infer harmonization and beat-alignment techniques on its own. During training, chord, vocal pitch contour, rhythm, key, structure can be directly extracted from the audio. During inference, we take advantage of the generated vocal midi track, where we can extract rhythm (beat and downbeat) and also do melody harmonization to obtain chord condition, referred to Figure 2. Additionally, we have add Section 2,3,4,5 in the demo page (https://composerflow.github.io/web/), showing the editability of the proposed pipeline.
>
> >A well-known issue with cascaded pipelines is the problem of cascading errors. An awkward melody from the first stage will lead to poor chords, which in turn will result in a subpar accompaniment. The paper does not address how ComposerFlow mitigates this.
>
> We agree that the cascading error is a problem cause by the lyrics-to-melody model. However, here are the efforts we made to prevent the cascading error:
> - Lyrics-to-melody: We set up a reference attribute set and a selection method described in Section 3.1. The reference statistical attributes will constrain the lyrics-to-melody generation, preventing awkward results.
> - Melody harmonization: Accomontage2 uses a chord progression matching paradigm to harmonize the given melody, and the chord templates from the chord library are either 4 bars or 8 bars, thus the chord progression itself will never be awkward.
> - Singing voice: We compare the register
> of the melody to typical vocal ranges—male (lower) vs. female (higher)—and test octave shifts∆ ∈ {−12, 0, +12} to align the melody with each profile’s comfortable tessitura. By doing these, we ensure the best quality of the singing voice.
> - Accompaniment generation: With the appropriate chord progression, singing voice, rhythm and other user given time-varying conditions, the backing generation will generate backing track as expected.
>
> >Furthermore, by breaking the process into isolated steps, the system loses the ability to reason about the song holistically. For example, the accompaniment model has no direct knowledge of the lyrics' sentiment, only the derived melody and chords. An end-to-end model, in theory, can learn these complex, high-level dependencies between lyrics, melody, harmony, and instrumentation. This work fails to discuss this fundamental trade-off.
>
> Due to the cascading pipeline, users can use text descriptions for SAG that fits the lyrics' and the singer's sentiment, for example, if using minor key to generate melody, then use text descritions with a sad mood for SAG. In theory, end-to-end model can holistically learns all the components, however, all open-source end-to-end song generation failed to removed copyright data, generating the song __holistically__ will infringe copyright issue. In contrast, __ComposerFlow__ use a staged generation method with licensed singers and only aims to learn to follow given time-varing conditions partially mitigates the infrigment issue.

---

> > ### Author Response · Authors · 2025-11-21
> >
> > >The evaluation, while extensive, has some notable issues: Comparing a Pipeline to End-to-End Models: It is fundamentally difficult to make a fair comparison between a modular system and a fully end-to-end one. ComposerFlow's superior lyric alignment (Table 3) is almost a given, as it uses a dedicated SVS module that is explicitly designed for this. This isn't a fair "win" but rather a direct consequence of the system's architecture
> >
> > __ComposerFlow__ is a staged pipeline that leverages multiple small modules. One of the reasons we use design in this way is because we can utilize the well-developed singing voice generation technique.
> >
> > >The subjective results (Table 5) show that while ComposerFlow is competitive among open-source models, it is significantly outperformed by the commercial system Suno across all metrics. Even among open-source models, it does not consistently lead, trailing AceStep in Overall Preference and Musicality. Its main strength is Voice Naturalness, which is again expected due to the dedicated SVS module. Overall, the generated music quality appears to be average at best.
> >
> > We use significantly less training data; for example, when comparing with Acestep, we use only 6k hours of data, while Acestep used 100k hours of training data. For compute resources, we used only a single RTX 3090, while they used 120 A100. The fact that we can use the SVS module is due to the design of ComposerFlow. Yielding an average result in all baselines should be satisfying when using significantly less training data.
> >
> > >The reported inference time of 55.5s (Table 3) is not particularly fast, especially compared to AceStep (13.0s) and DiffRhythm (11.5s). While much faster than autoregressive models like Levo, it doesn't represent a clear advantage in efficiency at inference time.
> >
> > We didn't claim our model as a fast-inference model. Our audio continuation method is a combination of an autoregressive and a non-autoregressive method, and slower than the non-autoregressive method is reasonable. However, our model can generate at most 47 seconds in each generation. To speed up, it is possible to generate multiple structure segments rather than one at a time, as long as the structure segments are not more than 40 seconds (leave 7 seconds as reference audio to ensure the quality).
> >
> > >The system is trained on Mandarin pop and evaluated on lyrics generated by ChatGPT. It is unclear how well this pipeline would generalize to other languages, genres (e.g., rap, where rhythm is paramount), or more poetic and structurally complex lyrics.
> >
> > One constraint of our model currently is that the lyrics-to-melody model only supports Chinese; thus, we can only generate Chinese pop songs.
> >
> > >The reliance on licensed data for the SVS, while ethically sound, also limits the vocal diversity of the system.
> >
> > That's true, but we deem it important to use licensed data at least for the singing voices. Vocal diversity can be improved when using more singing data, but we think that can be done in future work within the proposed pipeline. The main goal of the paper is to prototype and validate the core idea of the compositional song generation framework, and we are happy that our small-scale data helps to build a functional song generation model.

---

> > > ### Author Response · Authors · 2025-11-21
> > >
> > > >The issue of cascading errors is critical in a pipeline like this. Did you observe instances where a poor output from an early module (e.g., a monotonous melody from CSL-L2M) led to a complete failure in the final output? Are there any mechanisms in place, or that you envision, to handle this, such as a feedback loop or a joint optimization strategy?
> > >
> > > As described above,  the cascading errors are regulated by several methods, and rarely happens that led to a complete failure in the final output. The feedback loop the reviewer mentioned could be done by users, e.g. Edit unsatisfying melodies and chords, key, or even changing singers.
> > >
> > > >The SAG module is conditioned on several derived features (melody, chords, rhythm) but not directly on the raw vocals or lyrics. Do you think this limits the expressiveness of the accompaniment? For example, could an end-to-end model generate a more "empathetic" backing track that responds to the semantic meaning of the lyrics, which your system cannot?
> > >
> > > The alignment of the semantic between lyrics and backing can be regulated by using text prompts that semantically align the lyrics. As in traditional music production, the arrangement of the backing track is often done separately from the vocal track.
> > > Additionally, we evaluate MuseControlLite trained with different condition signals and the results are shown in Table 7. The result demonstrates that using raw vocal as condition leads to lower rhythm alignment and key accuracy with the vocal track.
> > >
> > > >In the subjective evaluation, what was the feedback from the "music experts"? Did they comment on the musical coherence and structure of the generated songs? The objective SongEval scores (Table 4) for ComposerFlow are decent but not outstanding. Does this align with expert opinion on aspects like harmonic progression and musical tension/release?
> > >
> > >
> > > >You mention that the pipeline is a "flexible paradigm rather than a fixed system." Could you elaborate on the challenges of swapping out a component? For example, if you were to replace the CSL-L2M melody generator with a different model that produces melodies in a different style, how much of the downstream pipeline (harmonization, accompaniment) would need to be re-tuned or re-trained to maintain quality?
> > >
> > > It is indeed easy to replace the component models by others in the proposed pipeline, as long as the input and output for each model are the same. For CSL-L2M, if the input are lrics with structure tags (just like other song generation models), output is vocal MIDI track (melody). Generally, SVS models use lyrics and vocal MIDI track (melody) as input, and generates singing voice. The proposed MIDI-SAG model require the vocal MIDI track to obtain rhythm and further retrieve chord with Accomontage, vocal pitch contour extracted from singing voice, structure and key assigned by users (same as the input for CSL-L2M). We update the demo page with Section 4, demonstrating an example of replacing CSL-L2M with SongComposer. However, the current released model of SongComposer can not generate whole song, which trim the input lyrics, and the quality of the generated melody is also worse than CSL-L2M.
> > >
> > >
> > > >Why was MuseControlLite chosen for the SAG task over other controllable generation models? And could you provide more detail on the decision to unfreeze the self-attention blocks of the Stable-Audio-Open backbone, and what effect this had on generation quality and continuity?
> > >
> > >
> > > - SingSong suffers from slow inference and is closed-source. FastSAG is limited to 10 seconds and does not support audio continuation. Additionally, conditioning on raw vocals may fail when the vocal track is silent (e.g., intro, outro, solo). Moreover, learning to follow the time-varying conditions designed in ComposerFlow is far more efficient than requiring the model to infer harmonization and beat alignment techniques on its own.
> > > - Informal listening indicates that audio continuation is smoother when the self-attention layers of the pretrained Stable-Audio Open are unfrozen. We conduct an evaluation of the audio continuation results comparing models with and without unfreezing the self-attention layers. The results are shown in Table 8. The FD, KL, CLAP are from [stable-audio-metrics](https://github.com/Stability-AI/stable-audio-metrics), and the Smoothness value proposed in MuseControlLite. For metrics from [stable-audio-metrics](https://github.com/Stability-AI/stable-audio-metrics) fine-tuning the self-attention effectively enhance the performance. However, the Smoothness value indicates opposite results from our perception, we consider this metrics not aligned with human perception.

---

### Official Review · Reviewer_qJGM · 2025-11-01

**Soundness:** 3
**Presentation:** 2
**Contribution:** 1
**Rating:** 2
**Confidence:** 4

**Summary:**

The paper introduces ComposerFlow, a modular, step-by-step pipeline for song generation that decomposes the process into four stages: (1) lyrics-to-melody generation, (2) melody harmonization, (3) singing voice synthesis (SVS), and (4) singing accompaniment generation (SAG). Unlike large end-to-end systems (e.g., Suno, Levo, DiffRhythm), ComposerFlow is designed to be lightweight and editable, trained entirely on a single RTX 3090 GPU with ~6k hours of data.

**Strengths:**

1. Resource Efficiency: Demonstrating that competitive quality can be achieved using only a single RTX 3090 GPU is a strong practical contribution, especially when compared to the massive compute requirements of end-to-end baselines. The “eco-friendly” framing is credible and backed by comparative data.

2. Systematic Evaluation: The inclusion of multiple objective metrics and a human listening study reflects a commendable level of rigor. The evaluation pipeline (with Whisper, CLAP, Audiobox, and SongEval) is comprehensive and reproducible.

**Weaknesses:**

1. Limited Technical Novelty: While the pipeline design is conceptually valuable, most components rely on existing models (e.g., CSL-L2M, AccoMontage2, MuseControlLite, FastSpeech). The contribution is primarily in integration and engineering, rather than novel model architectures or learning methods. For a venue like ICLR, this may raise questions about the depth of algorithmic innovation.

2. Weakness in Long-Form Consistency: The paper acknowledges that generated melodies deteriorate beyond 120 seconds, due to limited training length. This limits applicability for full-length songs or professional production contexts. The proposed workaround (audio continuation) only partially mitigates this issue.

3. Over-Reliance on Existing Controls: The pipeline’s success depends heavily on the quality of control signals (chords, rhythm, structure) extracted from pre-trained models like All-In-One and key-CNN. This creates a potential cascading error problem—small upstream mistakes (e.g., in chord detection) can degrade the final accompaniment. This limitation is not empirically analyzed.

**Questions:**

The training data used for ComposerFlow is relatively small at 6k hours, which is significantly lower than many of the baselines. However, is this training data sufficiently representative of the target domain, especially considering different musical genres and languages? Do you expect ComposerFlow to generalize well to other music styles or languages (e.g., English pop music or non-Western music)?

---

> ### Author Response · Authors · 2025-11-21
>
> Thank you for your comments!
>
> > Limited Technical Novelty: While the pipeline design is conceptually valuable, most components rely on existing models (e.g., CSL-L2M, AccoMontage2, MuseControlLite, FastSpeech). The contribution is primarily in integration and engineering, rather than novel model architectures or learning methods. For a venue like ICLR, this may raise questions about the depth of algorithmic innovation.
>
> We acknowledge that the writing of the original paper does not effectively make the technical novelty of our work clear. We have heavily rewritten our paper to have a focused presentation of the novel parts of our work. In our revision, we position our work as a response to a gap in existing end-to-end song generation systems. Although the idea of chaining specialized models is not new, prior work rarely measures how such pipelines actually compare—both in performance and computational efficiency—to strong end-to-end baselines, and usually treats each module as if it were independent. Our viewpoint is that this modular perspective is too naive: to get real gains in data efficiency and perceptual quality, the components must be co-designed and tightly integrated rather than simply concatenated.
> Concretely, our main technical novelty is MIDI-informed Singing Accompaniment Generation (MIDI-SAG), a new conditioning scheme for accompaniment models. Instead of relying only on raw vocal audio, we reuse information already produced in earlier stages of the pipeline—the vocal MIDI melody and a harmonization-derived chord progression—and feed these as conditions to the accompaniment generator. This lets us more reliably align beats and downbeats (via MIDI) and enforce harmonic coherence between vocals and backing (via chords), which is not possible from audio alone. We also tackle a practical but often overlooked issue: real songs contain instrumental sections with no vocals (intro, bridge, outro), where conventional SAG models “lose” their conditioning signal. We resolve this by leveraging the global rhythmic and harmonic structure of the full song and applying latent diffusion inpainting/outpainting to fill and extend these sections, yielding structurally complete, musically consistent full-track generations within a compositional framework.
>
> >Weakness in Long-Form Consistency: The paper acknowledges that generated melodies deteriorate beyond 120 seconds, due to limited training length. This limits applicability for full-length songs or professional production contexts. The proposed workaround (audio continuation) only partially mitigates this issue.
>
> This is true, but it's a constraint set by the off-the-shelf CSL-L2M model we use. We have now made it clear in the revised paper that the selection of specific model architectures throughout this paper is primarily for prototyping and validating the core idea of the compositional song generation framework. When a full-song-level, multilingual lyrics–melody model becomes available, the proposed MIDI-SAG model can accordingly perform full-song accompaniment generation.

---

> > ### Author Response · Authors · 2025-11-21
> >
> > >Over-Reliance on Existing Controls: The pipeline’s success depends heavily on the quality of control signals (chords, rhythm, structure) extracted from pre-trained models like All-In-One and key-CNN. This creates a potential cascading error problem—small upstream mistakes (e.g., in chord detection) can degrade the final accompaniment. This limitation is not empirically analyzed.
> >
> > End-to-end models also partially rely on existing controls. For instance, AceStep uses structure conditions from All-in-One, key from Essentia, and BPM from BeatThis. While these music information retrieval papers have already evaluated the effectiveness of their models, we would like to offer another point of view. The ablation study shown in Table 6 uses BeatNet rather than All-in-One to detect beat and downbeat, which means the conditions used for inference in MuseControlLite and the conditions used for evaluation rely on different beat detection techniques. However, when all conditions are given, the Rhythm F1 score is still very high, suggesting that the two beat detection models may have a consensus.
> >
> > Additionally, since key is highly related to chord progression, in Table 6 we observe that when the key condition is excluded, the model is still able to generate music with the same key accuracy as when all conditions are given, indicating that the model successfully follows the correct chord progression. However, when the chord condition is excluded, the model fails to follow the key condition. This indicates that the model fails to depend on the key condition alone, and that key accuracy is in fact dependent on the chord condition.
> >
> > >The training data used for ComposerFlow is relatively small at 6k hours, which is significantly lower than many of the baselines. However, is this training data sufficiently representative of the target domain, especially considering different musical genres and languages? Do you expect ComposerFlow to generalize well to other music styles or languages (e.g., English pop music or non-Western music)?
> >
> > Although the training data only contains Chinese pop (for CSL-L2M, SVS, and MIDI-informed SAG; Accomontage2 uses a chord template matching method, and its reference dataset is pop music chord progressions), the fine-tuning with MuseControlLite is primarily for learning to follow the given time-varying conditions. Thus, the fine-tuned MuseControlLite can leverage the knowledge of its pretrained backbone. Stable-Audio Open is trained on 7.3k hours of licensed data from Freesound and Free Music Archive, across different instruments and genres.
> > We show that MuseControlLite is capable of generating music that adheres to the given text description. The CLAP score reported in Table 3 shows comparable results to DiffRhythm and AceStep.

---

### Author Response · Authors · 2025-11-21
**For all reviewer (1)**

1. About the novelty of the paper: We acknowledge that the writing of the original paper does not effectively make the technical novelty of our work clear.  We have heavily rewritten our paper to have a focused presentation of the novel parts of our work.  In our revision, we position our work as a response to a gap in existing end-to-end song generation systems. We agree that the idea of chaining specialized models is not new, but we note that prior work rarely measures how such pipelines actually compare—both in performance and computational efficiency—to strong end-to-end baselines, and usually treats each module as if it were independent. Our viewpoint is that this modular perspective is too naive: to get real gains in data efficiency and perceptual quality, the components must be co-designed and tightly integrated rather than simply concatenated.
This leads to the main technical contribution of our work, the introduction of a new mechanism named MIDI-informed Singing Accompaniment Generation (MIDI-SAG).  It is a new conditioning scheme for accompaniment models. Instead of relying only on raw vocal audio, we reuse information already produced in earlier stages of the pipeline—the vocal MIDI melody and a harmonization-derived chord progression—and feed these as conditions to the accompaniment generator. This lets us more reliably align beats and downbeats (via MIDI) and enforce harmonic coherence between vocals and backing (via chords), which is not possible from audio alone. We also tackle a practical but often overlooked issue: real songs contain instrumental sections with no vocals (intro, bridge, outro), where conventional SAG models “lose” their conditioning signal. We resolve this by leveraging the global rhythmic and harmonic structure of the full song and applying latent diffusion inpainting/outpainting to fill and extend these sections, yielding structurally complete, musically consistent full-track generations within a compositional framework.
2. About the limitations of this paper: We have added the following paragraph to the paper to clarify this.
```We note that the selection of specific model architectures throughout this paper is primarily for prototyping and validating the core idea of the compositional song generation framework. In principle, any other suitable model architecture could be used. However, the specific choices impose practical constraints; for instance, as CSL-L2M currently supports only Chinese lyrics, our implementation is consequently focused on Mandarin singers in  SVS component and Mandarin pop for MIDI-SAG.```  When a full-song-level, multilingual lyrics–melody model becomes available, the proposed MIDI-SAG model can accordingly perform full-song accompaniment generation.

---

> ### Author Response · Authors · 2025-11-21
> **For all reviewer (2)**
>
> 3. About the cascading error: There can be indeed cascading errors, but we have actually made efforts to prevent them:
>     - Lyrics-to-melody: We set up a reference attribute set and a selection method described in Section 4.1. The reference statistical attributes constrain the lyrics-to-melody generation, preventing awkward results.
>     - Singing voice: We compare the register of the melody to typical vocal ranges—male (lower) vs. female (higher)—and test octave shifts Δ ∈ {−12, 0, +12} to align the melody with each profile’s comfortable tessitura. By doing this, we ensure the best possible quality of the singing voice.
>     - Accompaniment generation: With the appropriate chord progression, singing voice, rhythm, and other user-given time-varying conditions, the MIDI-informed SAG generates the backing track as expected.
>     - The melody harmonization model Accomontage2 uses a chord progression matching paradigm to harmonize the given melody, and the chord templates from the chord library are either 4 bars or 8 bars long. Thus, the chord progression itself tends to be good.
> - Actually, in our implementation, the cascading error seldom happens. The performance of the final output (song) are evaluated in Tables 4 and 5, showing average performance among the baselines.
> 4. Demonstraiting editablity: We update the demo page (https://composerflow.github.io/web/) with:
>     - Editing chord progressions and local text for each song structure segment.
>     - Same lyrics and melody, but edit the key and singer
>     - Same Lyrics and singer, different lyrics-to-melody model
>     - Demonstrate the same lyrics, but with different melodies.
>     - Table 6 shows the rhythm, chord, key alignment between the given condition and the generated music, results indicate that the MIDI-SAG highly adheres the given condition, which means the minor edits of the intermidiate results (e.g. chords) will be reflected.
> 5. Summary of the modification made during rebuttal:
>     -  Highlight MIDI-SAG vs. conventional SAG as the main technical contribution.
>     - Emphasize the structure-aware design that solves the problem when missing vocal tracks and achieves the structure completeness required for long-form generation.
>     - Add experiments comparing MIDI-SAG and conventional SAG.
>     - Add an ablation study for MIDI-SAG
>     - Add experiments showing the importance of unfreezing the self attention layers of Stable-Audio-Open.
>     - Update the demo page showcasing editability.

---

### Meta-Review · Area_Chair_Mwc1 · 2025-12-30

**Summary:**

All reviewers recommended to reject the paper (score 2). The main concerns were: (1) limited technical novelty (qJGM, yCCj, 7aJG); (2) lack of demonstration of editability, which is presented as an inherent advantage of the approach (7aJG, TYbL); sub-par quality compared to existing state of the art models, including an inherently limited long-form consistency (7aJG, TYbL, yCCj, qJGM); sensitivity to cascading errors and inheritance of the limitations of the weakest component in the cascade (qJGM, yCCj, 7aJG, TYbL); inaccurate framing of the method as resource efficient at training, and relatively slow inference (yCCj, 7aJG).

In their response, the authors emphasized that one of the main contributions of the paper is a MIDI-informed Singing Accompaniment Generation (MIDI-SAG) module. They additionally added editing demonstrations, reported training resources for all components combined, and explained why cascading errors are rare.

The AC appreciates the authors’ efforts, but still believes that the paper mostly presents an engineering contribution rather than a novel fundamental concept. In addition, the AC agrees with reviewer TYbL regarding the quality and musicality of the results in the demo page.

**Reviewer Concerns:**

Most major concerns have been addressed and clarified by the rebuttal. However, the AC still views the technical novelty as limited for an ICLR submission.

**Reviewer Scores:**

**qJGM: score 2.**

This is also the original score. The issue of technical novelty would likely still concern the reviewer.

**yCCj: score 2.**

This is also the original score. The issue of technical novelty would likely still concern the reviewer.

**7aJG: score 2.**

This is also the original score. The issue of technical novelty would likely still concern the reviewer.

**TYbL: score 4.**

The initial score was 2. The reviewer is predicted to have slightly raised the score thanks to the addition of editing illustrations. However, the concern of quality (including in the added editing experiments) is likely to have persisted.

---

### Decision · Program_Chairs · 2026-01-26

Reject